# Sample Average Approximation for Black-Box Variational Inference

**Javier Burroni**[1]        **Justin Domke**[1]        **Daniel Sheldon**[1]

[1] University of Massachusetts Amherst

## Abstract

Black-box variational inference (BBVI) is a general-purpose approximate inference approach that converts inference to a stochastic optimization problem. However, the difficulty of solving the BBVI optimization problem reliably and robustly using stochastic gradient methods has limited its applicability. We present a novel optimization approach for BBVI using the sample average approximation (SAA). SAA converts stochastic problems to deterministic ones by optimizing over a fixed random sample, which enables optimization tools such as quasi-Newton methods and line search that bypass the difficulties faced by stochastic gradient methods. We design an approach called "SAA for VI" that solves a sequence of SAA problems with increasing sample sizes to reliably and robustly solve BBVI problems without problem-specific tuning. We focus on quasi-Newton methods, which are well suited to problems with up to hundreds of latent variables. Our experiments show that SAA for VI simplifies the VI problem and achieves faster performance than existing methods.

## 1 INTRODUCTION

It is a long-standing research direction to develop robust inference methods that perform well on a wide range of real models. This is of immense practical interest in fields like astrophysics, epidemiology, political science, psychology, ecology, and others, where a scientist supplies a model and data, and the goal is to recover the posterior distribution of latent variables. However, inference is extremely challenging in general and formally intractable except for restricted cases, so approximations are needed.

Variational inference (VI) is one of the main approximate inference approaches. It poses inference as an optimization problem to find a distribution from a specified family that is as close as possible to the posterior by maximizing the evidence lower bound (ELBO) (Wainwright and Jordan, 2008; Jaakkola and Jordan, 1997; Beal, 2003), or, equivalently, minimizing the KL-divergence to the posterior.

In the quest to make VI broadly applicable and "automatic", recent work has focused on "black box" variational inference (BBVI) (Ranganath et al., 2014; Titsias and Lázaro-Gredilla, 2014; Kucukelbir et al., 2017; Yin and Zhou, 2018; Hoffman and Ma, 2020; Buchholz et al., 2018). BBVI performs ELBO maximization using only "black box" access to the model in the form of evaluations of the log joint density or the gradient thereof. This allows VI to be applied to a wide range of models, especially when paired with recent modeling frameworks such as Stan (Carpenter et al., 2017) that make it easy for users to specify models that are converted to routines for log-densities and gradients.

To achieve this generality, BBVI treats ELBO maximization as a stochastic optimization problem, which it solves via stochastic gradient descent (SGD) (Wingate and Weber, 2013; Blei et al., 2017; Kucukelbir et al., 2017; Ranganath et al., 2014; Rezende et al., 2014; Kingma and Welling, 2013) or a variant such as Adam (Kingma and Ba, 2015) or AdaGrad (Duchi et al., 2011). However, in practice, the difficulty of solving this stochastic optimization problem reliably and robustly has severely limited the applicability of BBVI (Agrawal et al., 2020; Welandawe et al., 2022). A particular challenge is selecting step size sequences that allow rapid progress and avoid suboptimality. This motivates the consideration of alternate stochastic optimization methods that can perform more reliably for BBVI problems.

In this paper, we propose an alternative optimization approach for BBVI based on the on sample average approximation (SAA) (Healy and Schruben, 1991; Robinson, 1996; Shapiro and Wardi, 1996; Kleywegt et al., 2002; Kim et al., 2015). A key feature of SAA is that it draws a fixed random sample and then solves a *deterministic* optimization problem. This enables tools such as line-search and second-

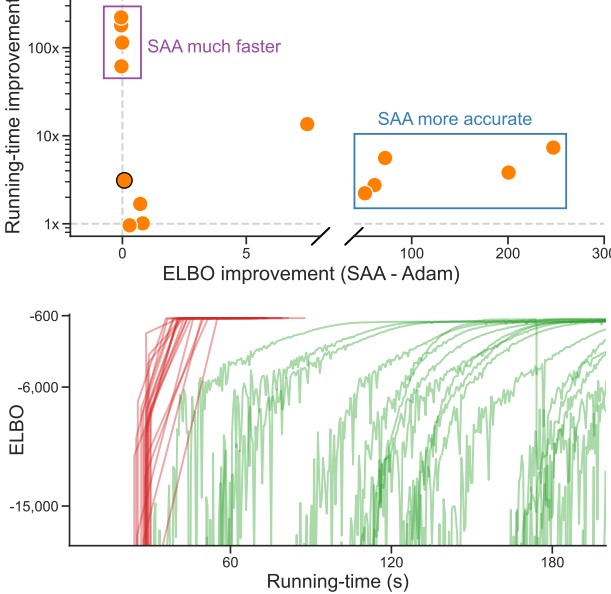

Figure 1: Top: ELBO improvement (nats) vs. running-time improvement (number of times faster) for SAA for VI compared to Adam, across 9 Stan models and 6 Bayesian logistic regression models using a dense-covariance Gaussian distribution. The bordered point ⬤ indicates that the models "australian" and "ionosphere" share the same coordinates. Bottom: Optimization traces for the "electric" model. See Section 5 and Appendix B for details.

order optimization, which are traditionally unavailable for BBVI but can substantially improve performance. We focus on the application of quasi-Newton methods with line search to BBVI with Gaussian approximating families. This is well suited to problems with up to several hundred latent variables, which covers a very large number of applied statistical models such as those that appear in the Stan model library, many of which remain very challenging for BBVI. Quasi-Newton SAA can also scale to much larger models when using diagonal Gaussian approximating families.

Figure 1 illustrates the speed and accuracy benefits of SAA compared to Adam when approximating the posterior distributions of 9 real Stan models and 6 Bayesian logistic regression models (see Table 2) using Gaussian distributions with dense covariance matrices. SAA is always comparable to or better to Adam in terms of solution quality, and, for 10 out of 15 models, either achieves a much better solution, or achieves a comparable solution much faster. Notably, nearly a third of the models are failure cases for Adam, where SAA finds a solution that is hundreds of nats better.

To achieve this robustness, we design the SAA for VI algorithm, which applies SAA to BBVI in an efficient and automatic way whenever the approximating family is reparameterizable. To address the Monte Carlo error introduced

by using a fixed random sample within SAA, we adapt techniques from the SAA literature to solve a sequence of problems with increasing sample sizes until a stopping criterion is reached (Chen and Schmeiser, 2001) and develop a custom stopping criterion for BBVI as well as default schedules for samples sizes and optimization tolerances to achieve robust out-of-the-box performance. SAA for VI also leverages the GPU-friendly nature of the SAA objective to increase optimization efficiency.

Our empirical results demonstrate that SAA for VI on our benchmark is competitive with state-of-the-art BBVI optimization methods—including first-order methods (Adam and AdaGrad) as well as a prior second-order stochastic optimization algorithm for BBVI (Liu and Owen, 2021)—while simplifying the variational inference process.

Concurrently with our work, Giordano et al. (2023) proposed a sample average approximation algorithm for variational inference, motivated by the same challenges of stochastic gradient methods that limit the robustness and broad applicability of BBVI. We discuss the relationship between our method and theirs in Section 4.

## 2 BACKGROUND

We are interested in approximating the posterior distribution of a latent variable given some observed data, i.e., $p(Z \mid x)$, where $Z$ is the latent variable and $x$ is the observed data. To achieve this, we will approximate the posterior with a distribution from an indexed family of approximations $\mathcal{Q} = \{q_\theta \mid \theta \in \mathbb{R}^d\}$, where $\theta$ is a vector of parameters that parameterize the approximation $q_\theta(Z)$, and $d$ is the dimension of $\theta$.

VI proposes to approximate the posterior distribution by finding a member from $\mathcal{Q}$ that is closest in Kullback-Leibler divergence to the true distribution. This is achieved by maximizing the evidence lower bound (ELBO), which is a function of the parameters:

$$\mathcal{L}(\theta) = \mathbb{E}[\ln p(Z, x) - \ln q_\theta(Z)], \qquad Z \sim q_\theta. \quad (1)$$

The optimization problem can be formulated as:

$$\max_{\theta \in \Theta} \mathcal{L}(\theta) = \max_{\theta \in \Theta} \mathbb{E}[\ln p(Z, x) - \ln q_\theta(Z)], \qquad Z \sim q_\theta. \quad (2)$$

Under smoothness assumptions, black-box VI presents this problem as a smooth stochastic optimization problem (SOP) and suggests solving it using methods based on stochastic gradient descent (SGD). Specifically, it uses stochastic gradient ascent to maximize the ELBO by updating the parameters as follows:

At every iteration $t$, samples $z_1, \ldots, z_n$ from $q_{\theta_t}$ are drawn and the sample mean of the function $g_{\theta_t}(Z)$ is being computed, where $g_{\theta_t}(Z)$ is a $\mathbb{R}^d$-valued random vector whose expectation equals the gradient. Then, this estimate is used,

along with some $\gamma_t \in \mathbb{R}_+$, to update the parameters according to:

$$\theta_{t+1} \leftarrow \theta_t + \gamma_t \frac{1}{n} \sum_{i=1}^{n} g_{\theta_t}(z_i). \quad (3)$$

This function can be obtained using various methods, including the score function estimator (Wingate and Weber, 2013; Ranganath et al., 2014) or, if the distribution is reparameterizable, the 'reparameterization trick' (Kingma and Welling, 2013; Fu, 2006; Kingma and Welling, 2019; Rezende et al., 2014), among others. A random variable $Z$ comes from a reparameterizable distribution $q_\theta$ if there exist a $C^1$ function $z_\theta$ and a density $q_{\text{base}}$ such that $Z = z_\theta(\epsilon)$ for $\epsilon \sim q_{\text{base}}$. We refer to these $\epsilon$ values as noise. In such case, the stochastic optimization problem becomes

$$\max_{\theta \in \Theta} \mathcal{L}(\theta) = \max_{\theta \in \Theta} \mathbb{E}[\ln p(z_\theta(\epsilon), x) - \ln q_\theta(z_\theta(\epsilon))], \quad (4)$$

where $\epsilon \sim q_{\text{base}}$. It then follows that, at every step $t$ of the optimization, the update rule of Eq. (3) is

$$\theta_{t+1} \leftarrow \theta_t + \gamma_t \frac{1}{n} \sum_{i=1}^{n} g_{\theta_t}(z_{\theta_t}(\epsilon_{ti})), \qquad \epsilon_{ti} \sim q_{\text{base}}.$$

Despite its simplicity, the explanation above fails to convey the complexities of choosing hyperparameters, particularly the step size $\gamma_t$, also known as the learning rate. The user can opt to use a step size schedule $\gamma = (\gamma_t)_{t \in \mathbb{N}} \subset \mathbb{R}_+$ that meets the Robbins-Monro conditions ($\|\gamma\|_1 = \infty$ and $\|\gamma\|_2 < \infty$), which can lead to SGD converging at a critical point due to the use of unbiased estimators of the gradients (Robbins and Monro, 1951; Ranganath et al., 2014; Jankowiak and Obermeyer, 2018). However, the specific sequence of the schedule is not specified and different schedules may affect the speed of convergence differently [cf. Agrawal et al. (2020)]. Critically, the random nature of estimating the loss function and its gradient makes it impractical to use traditional line-search methods. Additionally, the choice of the number of samples $n$ drawn at each iteration can affect the optimization process, as a larger $n$ provides a more accurate gradient estimate but may increase the computational cost. Balancing this trade-off is an important aspect of algorithm design.

Moreover, controlling the variance of gradient estimates significantly influences the performance of the optimization algorithm, affecting stability and convergence properties, and further adding to the complexity of the problem. In this context, the choice of the gradient estimator $g_{\theta_t}$ is crucial. Instead of employing the naïve estimator by taking the average of the gradient of $\ln p(z_{\theta_t}(\epsilon)) - \ln q_{\theta_t}(z_{\theta_t}(\epsilon))$, one can consider alternative methods such as the sticking-the-landing estimator (Roeder et al., 2017) or, when the entropy term $\mathbb{H}_\theta = -\mathbb{E}[\ln q_{\theta_t}(z_{\theta_t}(\epsilon))]$ is available in closed form, estimating the gradients of $\mathbb{E}[\ln p(z_{\theta_t}(\epsilon))] + \mathbb{H}_\theta$. Although all these estimators are unbiased, they exhibit different variance behaviors, which can impact the optimization process.

To reduce the variance of the gradient estimator, control variates can also be applied (Ranganath et al., 2014; Geffner and Domke, 2018). These choices contribute to the overall complexity of choosing hyperparameters, step size schedules, and the number of samples.

# 3 METHODS

## 3.1 SAMPLE AVERAGE APPROXIMATION

The problem of ELBO maximization in the reparameterization setting of Eq. (4) is formulated as an SOP where the stochasticity comes from a fixed probability distribution, i.e., a probability distribution which does not depend on $\theta$. Furthermore, the function inside the expectation is a smooth function of the parameters $\theta$. Solutions to these problems can be approximated using the *sample average approximation* (SAA): a sample average over a *fixed sample* replaces the expectation, effectively transforming the SOP into a deterministic optimization problem.

We propose to use SAA for black-box VI. To use SAA, we take $n$ i.i.d. samples $\epsilon = \epsilon_1, \ldots, \epsilon_n$ from the distribution $q_{\text{base}}$ and define the deterministic *training objective* function

$$\hat{\mathcal{L}}_\epsilon : \theta \mapsto \frac{1}{n} \sum_{i=1}^{n} [\ln p(z_\theta(\epsilon_i), x) - \ln q_\theta(z_\theta(\epsilon_i))],$$

which is a function of $\theta$ alone.

Then, the optimization problem in Eq. (4) can be transformed into a deterministic optimization problem

$$\max_{\theta \in \Theta} \hat{\mathcal{L}}_\epsilon(\theta) = \max_{\theta \in \Theta} \frac{1}{n} \sum_{i=1}^{n} [\ln p(z_\theta(\epsilon_i), x) - \ln q_\theta(z_\theta(\epsilon_i))]$$

$$= \max_{\theta \in \Theta} \frac{1}{n} \sum_{i=1}^{n} v_\theta(\epsilon_i), \quad (5)$$

where $v_\theta(\epsilon_i) = \ln p(z_\theta(\epsilon_i), x) - \ln q_\theta(z_\theta(\epsilon_i))$ denote the *log-weights*, also known as *log-importance ratios*. Since the optimization is performed with the fixed set $\epsilon$, we refer to it as the training noise.

We want to recover the optimal parameters $\theta^*$ of $\hat{\mathcal{L}}_\epsilon$. In an unconstrained smooth optimization setting, we need to specify how to compute a search direction and a step size. For the search direction, we will use L-BFGS (Broyden, 1970; Fletcher, 2013; Goldfarb, 1970; Shanno, 1970; Nocedal, 1980).

In contrast to the SGD setting, deterministic optimization allows us to specify the step size using line search and ask for it to satisfy the *strong Wolfe conditions* (Nocedal and Wright, 1999). Specifically, for $0 < c_1 < c_2 < 1$, the step size $\gamma$ must simultaneously satisfy the modified curvature

(MC) and sufficient increase (SI) conditions, that is,

$$\left|\nabla \hat{\mathcal{L}}_{\boldsymbol{\epsilon}}(\theta + \gamma \mathbf{r})^{\mathrm{T}} \mathbf{r}\right| \leq c_2 \left|\nabla \hat{\mathcal{L}}_{\boldsymbol{\epsilon}}(\theta)^{\mathrm{T}} \mathbf{r}\right|, \qquad \text{and,} \qquad \text{(MC)}$$

$$\hat{\mathcal{L}}_{\boldsymbol{\epsilon}}(\theta + \gamma \mathbf{r}) \geq \hat{\mathcal{L}}_{\boldsymbol{\epsilon}}(\theta) + c_1 \gamma \nabla \hat{\mathcal{L}}_{\boldsymbol{\epsilon}}(\theta)^{\mathrm{T}} \mathbf{r}. \qquad \text{(SI)}$$

We will use L-BFGS with line search to find a local optimum of Eq. (5), and denote the process that does so by $\mathrm{Opt}(\theta, n, \boldsymbol{\epsilon}, \tau)$. Here, $\tau$ is the maximum number of iterations for which L-BFGS will run, and $\theta$ is an initial value of the parameters. Besides the arguments of $\hat{\mathcal{L}}_{\boldsymbol{\epsilon}}(\theta)$, we also need to specify the value of $\tau$.

**Sandwiching the optimal ELBO** Critically, the training objective $\hat{\mathcal{L}}_{\boldsymbol{\epsilon}}(\theta)$ and the ELBO $\mathcal{L}(\theta)$ may differ for a fixed $\theta$. The ELBO, as defined in Eq. (1), is an expectation over the distribution $q_\theta$, while the training objective is computed based on an average over a fixed sample $\boldsymbol{\epsilon}$. In contrast, the optimal ELBO refers to the value of the ELBO achieved by the maximizer of Eq. (2), denoted as $\theta^*$, and depends only on the target distribution and the approximating family.

During optimization with a fixed sample of training noise $\boldsymbol{\epsilon}_n = \epsilon_1, \dots, \epsilon_n$, one might wonder how much the learned parameters $\theta_{\boldsymbol{\epsilon}_n}^*$ and the distribution $q_{\theta_{\boldsymbol{\epsilon}_n}^*}$ depend on these noise samples, and, in particular, how this dependency translates into the tightness of the gap between the ELBO $\mathcal{L}(\theta_{\boldsymbol{\epsilon}_n}^*)$ of the learned approximation and its upper bound, the optimal ELBO $\mathcal{L}(\theta^*)$. Fortunately, two results by Mak et al. (1999) are relevant to our discussion. Note that until the noise variables $\epsilon_1, \dots, \epsilon_n$ are realized, the quantity $\theta_{\boldsymbol{\epsilon}_n}^*$ and all functions of it are random. Let $\hat{\boldsymbol{\epsilon}}_{n+1} = \hat{\epsilon}_1, \dots, \hat{\epsilon}_{n+1}$ be a sample of size $n+1$ taken i.i.d. from $q_{\mathrm{base}}$. Assuming the deterministic optimization with fixed noise converges to a global optimum, it holds that: (i) the ELBO and training objective sandwich the optimal ELBO (in expectation), that is, $\mathcal{L}(\theta_{\boldsymbol{\epsilon}_n}^*) \leq \mathcal{L}(\theta^*) \leq \mathbb{E}\,\hat{\mathcal{L}}_{\boldsymbol{\epsilon}_n}(\theta_{\boldsymbol{\epsilon}_n}^*)$; and (ii) the training objective converges monotonically to the optimal ELBO from above (in expectation), that is, $\mathbb{E}\,\hat{\mathcal{L}}_{\hat{\boldsymbol{\epsilon}}_{n+1}}(\theta_{\hat{\boldsymbol{\epsilon}}_{n+1}}^*) \leq \mathbb{E}\,\hat{\mathcal{L}}_{\boldsymbol{\epsilon}_n}(\theta_{\boldsymbol{\epsilon}_n}^*)$.

In particular, these results mean that we can use standard statistical techniques to quantify the discrepancy between the ELBO $\mathcal{L}(\theta_{\boldsymbol{\epsilon}_n}^*)$ and the training objective $\hat{\mathcal{L}}_{\boldsymbol{\epsilon}}(\theta_{\boldsymbol{\epsilon}_n}^*)$ by comparing the distribution of the log-weights $v_1, \dots, v_n$ for a fresh sample of noise, referred to as testing noise, and the training noise, a technique first used by Mak et al. (1999). Figure 2 displays the distribution of log-weights for a growing sample size. As the number of samples increases, the training objective value decreases and approaches that of the ELBO estimation, which in turn increases, indicating progress toward the ultimate goal of ELBO maximization, while tightening the gap around the optimal ELBO.

We adopt the classical approach of tightening this gap by solving a sequence of SAA approximations for an increasing sequence of sample sizes $(n_t)_{t \in \mathbb{N}} \subseteq \mathbb{N}$, which creates a sequence of solutions $(\theta_{n_t}^*)_{t \in \mathbb{N}}$. Shapiro (2003) give general

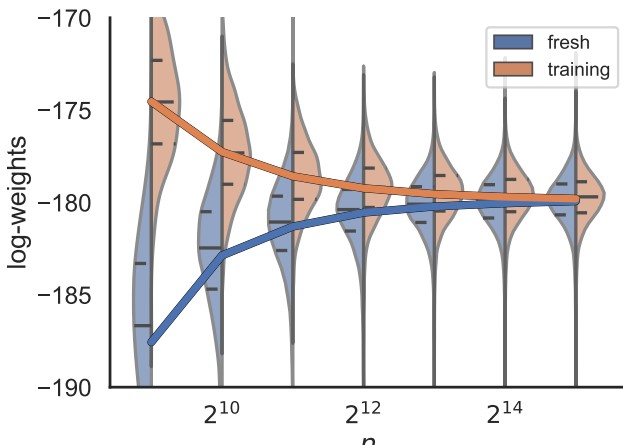

Figure 2: Distribution of log-weights as a function of optimization sample size $n$ (`mushrooms` dataset). The violin plot shows the distributions, with overlaid lines indicating means for both fresh and training samples. These means provide estimations for the ELBO and training objective.

conditions for the set of optimal solutions (or critical points) of SAA problems to converge to the corresponding set for the original stochastic optimization problem. The conditions include uniform convergence of the SAA objective functions and compactness of the solution set (see also Kim et al. 2015). While these could likely be applied to VI problems, the conditions, especially compactness of the solution set, would be problem specific and depend, for example, on the particular parameterization of a variational distribution, and we don't explore it further.

## 3.2 ALGORITHM

We present an algorithm that uses SAA to approximate the solution to the optimization problem of maximizing the ELBO. Our objective is to find a good approximation to the solution with a reasonable computational cost and reduce the gap between the ELBO and the training objective after optimization, as described above. To this end, we build our stopping criteria based on comparing distributions of log-weights. The algorithm, described in Algorithm 1, consists of two procedures: the optimizer $\mathrm{Opt}$ and the convergence checker. We previously described the optimizer, in which we used a quasi-Newton method. The convergence checker is a function that determines whether we need to continue the optimization process, and we will describe it later in this section.

The algorithm initializes with guess $\theta_0$, sample size $n_0$, and maximum optimizer iterations $\tau_0$. In each iteration $t$, we double the sample size to tighten the gap around the optimal ELBO. We draw training noise $\boldsymbol{\epsilon}_{n_t} = \epsilon_1, \dots, \epsilon_{n_t}$ from the base distribution $q_{\mathrm{base}}$ and then use the optimizer to find the maximizer $\theta_t^*$ of the deterministic objective $\hat{\mathcal{L}}_{\boldsymbol{\epsilon}_{n_t}}$. If the

optimizer reaches the iteration limit $\tau_t$, we double its value.

When the optimizer Opt finishes in a small number of iterations, the parameters may remain almost unchanged, resulting in nearly identical log-weights. Consequently, any convergence test based on these log-weights might not be indicative. Though such behavior could signal convergence, it might be due to chance. To address this uncertainty, we require a minimum of `VERY_SMALL_ITER` iterations before considering convergence. However, if the optimizer finishes without reaching this number of iterations for three consecutive step sizes, we stop the process.

---

**Algorithm 1** SAA for VI

---

1: **Input:** $\theta, n, \tau$        **Output:** parameters $\theta^*$
2: $t \leftarrow 0, \text{count} \leftarrow 0$
3: **while** count $< 3$ **do**
4:     $t \leftarrow t+1, n \leftarrow 2n$
5:     $\boldsymbol{\epsilon}_n \leftarrow \epsilon_1, \ldots, \epsilon_n,$           $\epsilon_i \sim q_{\text{base}}$
6:     $\theta \leftarrow \text{Opt}(\theta, n, \boldsymbol{\epsilon}_n, \tau)$
7:     $\eta \leftarrow$ number of iter used by the optimizer
8:     **if** $\eta = \tau$ **then**
9:        $\tau \leftarrow 2\tau$
10:    **if** $\eta <$ VERY_SMALL_ITER **then**
11:       count $\leftarrow$ count $+ 1$
12:    **else**
13:       count $\leftarrow 0$
14:    **if** count $= 0$ **and** converged?$(\theta, \boldsymbol{\epsilon}_n, t)$ **then**
15:       **break**
16: **return** $\theta^* \leftarrow \theta$

---

**Algorithm 2** converged?

---

1: **Input:** $\theta, \boldsymbol{\epsilon}_n, t$       **Output:** True if converged
2: $\hat{\boldsymbol{\epsilon}}_{10k} \leftarrow \hat{\epsilon}_1, \ldots, \hat{\epsilon}_{10k},$         $\hat{\epsilon}_i \sim q_{\text{base}}$
3: obj $\leftarrow \text{mean}(v_\theta(\boldsymbol{\epsilon}_n))$
4: elbo $\leftarrow \text{mean}(v_\theta(\hat{\boldsymbol{\epsilon}}_{10k}))$
    $\triangleright$ Statistically compare means:
5: $\text{p}_{\text{value}} \leftarrow \text{t\_test}(v_\theta(\boldsymbol{\epsilon}_n), v_\theta(\hat{\boldsymbol{\epsilon}}_{10k}))$
6: **if** $\text{p}_{\text{value}} > 0.01$ **then**
7:     **return** True
8: **if** $|\text{obj} - \text{elbo}| < \delta$ **or** $t \geq \text{max\_t}$ **then**
9:     **return** True
10: **return** False

---

**Stopping** Algorithm 2 defines the stopping criteria for our optimization process, which involves computing log-weights. Specifically, given the training noise $\boldsymbol{\epsilon}_{n_t}$ and the parameters $\theta_t$, we compute the log-weights $v_{\theta_t}(\epsilon_1), \ldots, v_{\theta_t}(\epsilon_{n_t})$, which we denote as $v_{\theta_t}(\boldsymbol{\epsilon}_{n_t})$. We also compute a new set of log-weights using 10k fresh samples of testing noise, denoted by $v_{\theta_t}(\hat{\boldsymbol{\epsilon}}_{10k})$.

To decide whether to halt or continue the optimization process, we use a two-sided t-test to compare the means of log-weights. We compare the mean log-weight calculated with the training noise, $v_{\theta_t}(\boldsymbol{\epsilon}_{n_t})$, to the mean log-weight computed from the testing noise, $v_{\theta_t}(\hat{\boldsymbol{\epsilon}}_{10k})$. The null hypothesis asserts that these means are the same. If we cannot reject the null hypothesis, we terminate the optimization process. Although the assumptions required for the t-test (e.g., that the training log-weights are i.i.d.) might not strictly hold in all cases, we employ this statistical test as a heuristic for stopping the optimization. Our approach draws inspiration from the methodology outlined in Mak et al. (1999). Alternatively, statistical tests such as the Kolmogorov-Smirnov or the Cramér-von Mises could be used to directly compare the log-weight distributions. In Appendix H, we evaluate the alternatives and show that the t-test is a reasonable choice.

Our optimization process terminates when the null hypothesis cannot be rejected with a significance level of $1\%$. Checking for convergence only when count $= 0$ avoids meaningless tests, as without optimizer updates, the distributions of training log-weights $v_{\theta_t}(\boldsymbol{\epsilon}_{n_t})$ and testing log-weights $v_{\theta_t}(\hat{\boldsymbol{\epsilon}}_{10k})$ would be nearly identical. We also introduce two additional stopping conditions: the maximum number of iterations $\text{max\_t}$ and the threshold $\delta$ for the difference between the training objective $\hat{\mathcal{L}}_{\boldsymbol{\epsilon}}(\theta_t)$ and the estimated ELBO $\hat{\mathcal{L}}_{\hat{\boldsymbol{\epsilon}}_{10k}}(\theta_t)$. In our experiments, we set $\text{max\_t}$ to ensure that the maximum sample size was $n_{\max} = 2^{18}$, and $\delta$ to 0.01. In Appendix F, we provide a more detailed discussion of the hyperparameters used in our experiments.

## 4 RELATED WORK

In the existing literature, there are efforts to incorporate second-order information into stochastic optimization, which have been applied to VI. Byrd et al. (2016) introduced a method that employs the L-BFGS update formula through subsampled Hessian-vector products, referred to as batched-L-BFGS or batched quasi-Newton. Liu and Owen (2021) applied the method from Byrd et al. (2016) to address the variational inference problem, with the optional inclusion of quasi-Monte Carlo (QMC) sampling to further decrease the variance of the gradient estimator. Both approaches involve a two-step algorithm: (1) updating the parameters at each iteration using L-BFGS's two-loop recursion, and (2) updating the displacement vector $\mathbf{s}$ and gradient difference vector $\mathbf{y}$ of L-BFGS every $B$ steps by employing the average of the parameters from the preceding $B$ iterations. In the work of Liu and Owen (2021), each iteration involves drawing a fixed-size sample of noise $\epsilon$ from $q_{\text{base}}$ to estimate the ELBO gradient and conduct the line search. The sample size is not extensively discussed in their work; however, the experiments were conducted with sample sizes of 128 or 256. These values are larger than those typically used in the literature, suggesting that the sample size could indeed be a

relevant factor to consider. Our method deviates from the approach proposed by Liu and Owen (2021) in two key ways. Firstly, we execute a complete deterministic optimization using a fixed set of noise, effectively reducing uncertainty. Secondly, we seamlessly integrate the sample size consideration into the algorithm itself, consequently minimizing the need for user input. As we demonstrate in Section 5.1.2, these differences lead to significant improvements when handling complex target and approximating distributions.

An alternative approach to incorporating second-order information into the variational inference problem can be found in the work of Zhang et al. (2022). Their method employs L-BFGS to identify modes or poles of the posterior distribution. Subsequently, the data generated by L-BFGS is utilized to estimate the posterior covariance around the mode, which is then used to parameterize an approximating distribution. This approach more closely resembles the Laplace approximation than methods that seek approximations to a global optimizer of the ELBO from a fixed parametric family.

We share a common goal with Welandawe et al. (2022), who also drew inspiration from Agrawal et al. (2020) to develop a system for variational inference that requires minimal user input. However, their method employs SGD for optimizing the ELBO and uses a heuristic schedule to update the step size $\gamma_t$ during the optimization process. They initially use a fixed step size and incorporate tools to detect when the SGD process reaches stationarity, at which point they decrease the step size by a factor $\rho$. During the stationary regime, they calculate the average of the parameters and take it as the optimal parameters for a given step size $\theta^*_{\gamma_t}$. They repeat the process of decreasing the step size until the symmetrized KL divergence between the current distribution and the optimal distribution $q_*$ (for the approximating family) falls below a threshold $\xi$. Notably, since the optimal distribution $q_*$ is not known, the authors estimated the KL divergence between $q_*$ and the current distribution $q_{\theta^*_{\gamma_t}}$. The authors observed that taking the average of the parameters in the stationary regime significantly improves the approximation quality compared to considering each parameter at every iteration.

In the machine learning literature, the application of sample average approximation has been relatively rare. Some early works include PEGASUS by Ng and Jordan (2000), in which the authors addressed partially observable Markov decision processes by replacing the *value of a policy* (an expectation) with the sample average of the value function applied to a finite number of states for optimization purposes. In a different context, Sheldon et al. (2010) explicitly utilized the sample average approximation technique in a network design setting, where a naïve greedy approach was not applicable. More recently, Balandat et al. (2020) adopted sample average approximation to optimize the acquisition function in Bayesian optimization. SAA was previously used for VI in a specialized capacity in several papers (Giordano et al., 2018; Domke and Sheldon, 2018; Giordano et al., 2019;

Domke and Sheldon, 2019; Giordano et al., 2022); our work and the concurrent work of Giordano et al. (2023) are the first to explore its general applicability.

As mentioned in the introduction, Giordano et al. (2023) concurrently and independently developed a method based on the sample average approximation for black-box variational inference. The two papers employ the same basic algorithmic idea but have several differences in scope. Unlike Giordano et al. (2023), we focus substantially on the case where SAA with a fixed sample size has significant error and therefore one needs to solve a sequence of problems with increasing sample sizes. We introduce heuristics that guide the selection of sample sizes and the decision of when to halt the process. On the other hand, Giordano et al. (2023) exploit the determinism of the SAA problem to develop techniques based on sensitivity analysis and the theory of "linear response covariances" (Giordano et al., 2015, 2018) to improve posterior covariance estimates of black-box VI and to estimate the Monte Carlo error of the SAA procedure, which are outside the scope of our work. They present a theoretical result indicating a failure mode for SAA when the number of samples is too few compared to the dimension of the latent variables. Specifically, for a Gaussian approximation with a dense covariance matrix, the sample size $n$ must be at least equal to the dimension $d_Z$ of the latent space for the SAA problem to be bounded. Interestingly, although they conclude that this limitation prevents the use of SAA for VI with a dense Gaussian approximation, we show in the experiments section that, for interesting models, it is indeed feasible. Two reasons for this discrepancy are: (1) our SAA sample sizes can reach up to $2^{18}$, unlike their usual size of 30, and (2) our largest model has 501 latent variables, whereas theirs have up to 15K. Thus, their theoretical result provides useful guidance on the limitations of SAA for VI, while our empirical work shows that SAA for VI can be practical up to quite large sample sizes. We have provided an addendum in Appendix E that uses their theoretical result to improve our method: when using a dense approximation, the sequence of SAA problems should begin with a sample size larger than $d_Z$; this makes SAA for VI even faster by avoiding wasted effort for small sample sizes.

Another related work is Gianniotis et al. (2016), who independently developed a variational inference approach that uses the reparameterization trick with Gaussian distributions and optimizes a Monte Carlo approximation to the ELBO, mirroring work in the machine learning community [cf. Kucukelbir et al. 2017; Kingma and Welling 2013; Ranganath et al. 2014]. Like us, Gianniotis et al. (2016) also optimize over a fixed set of noise samples and assess overfitting. However, there are several differences in focus. Most notably, our work is developed in the more general context of BBVI and applies to a wide range of models and approximating families, as long as they are reparameterizable.

We also frame our approach within the lens of stochastic optimization, which is consistent with our overall goal of using second-order information to solve the BBVI problem in a robust way that requires minimal user input.

## 5 EXPERIMENTS

We now present experiments comparing SAA to other methods in terms of optimization quality and running time. We examine two types of models following the setup of Burroni et al. (2023). These include 11 models from the Stan model library[1] (Stan Development Team, 2021; Carpenter et al., 2017) as well as Bayesian logistic regression models applied to 6 UCI datasets (Dua and Graff, 2017).[2] Details of the datasets are in Appendix A. For each model $p(\mathbf{Z}, x)$, where $\mathbf{Z}$ is a $d_Z$-dimensional random vector, the approximating distribution $q_\theta$ can either be a diagonal Gaussian or a $d_Z$-dimensional multivariate Gaussian distribution. The former is a product of $d_Z$ independent Gaussians, where the parameters $\mu_i$ and $\sigma_i^2 > 0$ are specific to each $Z_i$. The latter has parameters $\mu_i$ and $LL^{\mathrm{T}}$, where $L \in \mathbb{R}^{d_Z \times d_Z}$ is a lower-triangular matrix with diagonal elements that are positive, enforced by applying the `softplus` transformation.[3] We run all our experiments on GPUs.

We conduct performance comparisons and an ablation study. We compare primarily to Adam with a fixed step-size, which is commonly used for black-box VI optimization, and batched quasi-Newton, a newer method that introduces second-order information in the optimization process. We also compare to Adagrad. For all baseline methods, we use the naïve gradient estimator described in Section 2. When using Gaussian approximating distributions, this estimator corresponds to the one obtained when the entropy term is computed in closed-form. In the ablation study, we explore how our decisions affect the algorithm's performance.

### 5.1 PERFORMANCE COMPARISON

#### 5.1.1 Adam

Adam (Kingma and Ba, 2015) is a standard default optimizer for BBVI. Step size choice is less relevant with Adam than with SGD but still a factor to consider. For each experiment (combination of model and approximating family) we ran Adam with three different step-sizes (0.1, 0.01, and 0.001) and ran 20 repetitions of each combination. At each iteration, we estimated the gradient of the ELBO by taking 16 samples from $q_\theta$. For each model and approximating family, we

---

[1]congress, election88, election88Exp, electric, electric-one-pred, hepatitis, hiv-chr, irt, mesquite, radon, and wells

[2]a1a, australian, ionosphere, madelon, mushrooms, and sonar

[3]Following Kucukelbir et al. (2017), we transform the model $p$ into one with unconstrained real-valued latent variables, using PyTorch's (Paszke et al., 2019) constraints framework.

selected the step size in hindsight that provided the highest median ELBO across the 20 repetitions. (See Appendix B for more details on the Adam experiments.) For SAA for VI, we used the algorithm described in Section 3.2 with the default parameter values of Table 15 in the appendix.

We conducted two comparisons. First, we assessed the median ELBO, obtained across 20 repetitions, at the end of the optimization process using both Adam and SAA for VI. We initially ran Adam for $40,000$ iterations, but found that more iterations were needed for some models and increased the number for models such as election88, electric, irt, madelon, and radon; see details in the appendix.

In the second comparison, we focused on the time required to reach a specified ELBO. For each model and approximating distribution we identified as a "benchmark ELBO" the smaller of the two median ELBO values achieved by Adam and SAA, respectively, across their 20 repetitions. In other words, this ELBO value was achieved in at least half of the runs by both optimizers. We then evaluated how long it took for each method to reach an ELBO value within 1 nat of the benchmark ELBO.

Figure 1 and Table 1 show summary results for Stan models and Bayesian logistic regression with dense Gaussian distributions; detailed ELBO comparisons and additional models appear in Table 3 in the appendix. See also Figure 3, which compares the final ELBO values obtained by all methods evaluated. Although Adam occasionally attains marginally superior median ELBO values for certain models——due to the stopping criterion of SAA for VI——SAA for VI consistently achieves higher median ELBOs for complex models. We noticed that Adam's performance was erratic for models like election88Exp and had a tendency to diverge, especially for the hepatitis model when optimized beyond $40,000$ iterations. This divergence partially accounts for the pronounced disparity in median ELBOs between Adam and SAA for VI. We note that it's possible that Adam could achieve higher ELBO values by searching over a finer step-size grid; however, it is exactly this type of difficult and time-intensive tuning we seek to avoid with SAA. Table 4 in the appendix lists the time each method takes to achieve the adjusted ELBO and their respective ratios. SAA for VI is almost always faster, often by factors of 10 to 100. For instance, optimizing the electric model using Adam takes about a minute, whereas SAA for VI accomplishes the same in under 2 seconds, making SAA more than 30 times faster. Note that for Adam we only counted the compute time of the best-performing of the three learning rates, making the comparison even more favorable for SAA for VI. Since GPUs allow for vectorized multi-sample model evaluation, the wall clock time in seconds serves as the most meaningful metric for comparing the compute time of both methods. Given these results, we confidently conclude that SAA for VI is a faster alternative to Adam in these scenarios.

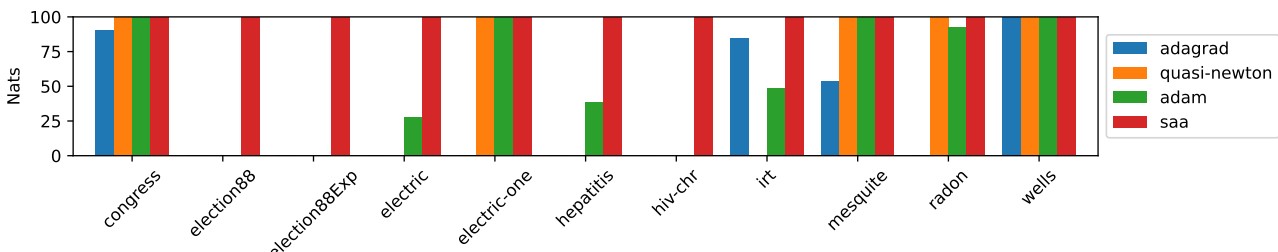

Figure 3: ELBO comparison of different methods on Stan models with dense Gaussian approximation. For each model, ELBOs are shifted so the best model has value 100, and methods more than 100 nats worse are not shown. Only SAA achieves robust performance across all models. For quasi-Newton, we choose the best performing sample size.

Appendix G provides additional results to explore the effect of different sample sizes (ranging from 1 to 256) for Adam as well as a different optimizer (Adagrad, Duchi et al. 2011; see also Figure 3). Across all settings, SAA for VI was consistently fast and robust compared to these alternatives.

Finally, to show that SAA for VI can also be effective in larger models, we learned an approximate posterior for a stochastic volatility model from Chib et al. (2009); see also (Naesseth et al., 2018; Lai et al., 2022). To make the task more challenging, we switched from monthly to daily data, increasing the data points processed and the number of latent variables. Since this model consists of 17,228 latent variables, using a dense covariance matrix would imply hundreds of millions of parameters, making the approach impractical. However, with a diagonal covariance matrix, SAA for VI finds a solution in less than 30 seconds, while Adam takes up to 2 minutes. (See Figure 4 in the appendix.)

### 5.1.2 Batched quasi-Newton

As noted in Section 4, our method differs from the batched quasi-Newton approach by Liu and Owen (2021), which also incorporates second-order information into VI. We now empirically show the impact of these differences, specifically the use of a sequence of sample average approximations with an increasing number of samples. We implemented the batched quasi-Newton method in PyTorch (without quasi-Monte Carlo sampling) and ran 20 independent runs of 40,000 iterations in each experiment. We started with a sample size of 16, then repeatedly doubled the number of samples up to a maximum of 128 for models where the method encountered difficulties. We set the update frequency $B$ (see Section 4) to 20 as recommended in the original paper.

With diagonal-covariance Gaussians, the batched quasi-Newton method shows performance on par with SAA for VI (Table 8 in the appendix). However it struggles significantly with dense Gaussians, and fails to find good solutions for many models, as shown in in Figure 3 and Table 9 in the appendix. The batched quasi-Newton method reaches optimal performance for most Bayesian logistic regression models

|  | Dense Covariance | | |
|---|---|---|---|
|  | SAA for VI | Adam | Impr. |
| **Stan models** | | | |
| congress | 423.55 | 423.58 | -0.03 |
| election88 | -1,398.03 | -1,645.18 | 247.15 |
| election88Exp | -1,381.79 | — | — |
| electric | -786.91 | -859.26 | 72.35 |
| electric-one-pred | -818.01 | -818.00 | 0.01 |
| hepatitis | -557.36 | -618.76 | 61.40 |
| hiv-chr | -582.78 | — | — |
| irt | -15,884.67 | -15,936.06 | 51.39 |
| mesquite | -29.83 | -29.78 | -0.05 |
| radon | -1,209.46 | -1,216.92 | 7.46 |
| wells | -2,041.95 | -2,041.90 | -0.05 |

Table 1: **ELBO** of SAA for VI and Adam for Stan models using a dense covariance matrix, highlighting the *improvements* in ELBO by SAA for VI over Adam. Various step sizes were explored for Adam, and the best results are reported. For additional datasets and approximating distributions, see Appendix B.

but faces difficulties with models from the Stan example library. Even with a sample size of 128, a significantly larger value than commonly employed with SGD, the method still falls short of the best ELBO values achieved by other methods. Additionally, the wall-clock time taken by the batched quasi-Newton method is often similar to or slower than the time taken by SAA for VI (Table 10 in the appendix).

### 5.2 ABLATION STUDY

**Impact of warm start.** The optimization process requires a decision on whether to use warm start or draw fresh parameters for each iteration. Once the inner optimization process $\text{Opt}(\cdot)$ converges to parameters $\theta_t^*$, it may still be necessary to increase the sample size and run it more times, as described in Section 3.2. Should we initialize the parameters

with $\theta_t^*$ or instead draw a new set of parameters?

Pasupathy (2010) provides an intuition of why using a warm start is helpful: in principle, the optimization process for larger sample sizes begin from a place that probably is close to a solution. To empirically verify this intuition, we conducted an experiment to compare the performance of warm start and drawing fresh parameters across different models and approximating distributions. For each combination of models and distribution, we ran the sequence of SAA problems until convergence, using either warm start or by sampling new parameters at the beginning of each inner optimization. Specifically, at each iteration $t$, we initialized the process either with the previously computed optimal parameters $\theta_{t-1}^*$ (warm start) or by drawing a new random set of parameters (fresh start). Our results, presented in Table 12 in the appendix, show that using warm start results in a significant reduction in the total time taken to converge. For example, on the election88 dataset, using fresh samples takes $20\times$ more time than using a warm start.

## 6 LIMITATIONS AND DIRECTIONS FOR EXTENSIONS

We presented the SAA for VI method under the assumption that the approximating family is reparameterizable. This assumption was explicit in the formulation of the stochastic optimization problem in Equation (4) and the deterministic optimization problem in Equation (5). While we focused on using Gaussian approximating families, the SAA for VI algorithm can be applied to other reparameterizable families, such as normalizing flows (Tabak and Turner, 2013; Rezende and Mohamed, 2015; Kingma et al., 2016; Papamakarios et al., 2021; Agrawal et al., 2020). An interesting direction for future work would be extending SAA for VI to non-reparameterizable families. In this context, a recent work by Zimmermann et al. (2024) proposed optimizing a forward KL divergence objective using the sample average approximation while removing the reparameterizable family restriction.

Two other limitations relate to scalability: (1) our method does not currently scale to models with very large numbers of latent variables unless diagonal Gaussian approximating families are used, and (2) SAA for VI using quasi-Newton methods does not support subsampling for models with large numbers of local latent variables. For (1), future work can consider extensions that enrich the variational family beyond diagonal Gaussians while retaining scalability of SAA, such as hierarchical distributions (Agrawal and Domke, 2021) or normal distributions with a "diagonal plus low-rank" covariance structure Tomczak et al. (2020). Other tools that increase the model capacity for the covariance matrix by slowly increasing the number of parameters, like the Householder flow (Tomczak and Welling, 2017), could also be considered. For (2), future work may consider alternative

optimizers for the deterministic subproblem of SAA for VI that support data subsampling while still benefiting from the deterministic fixing of parameters, such as first-order methods that exploit a finite-sum structure (Vaswani et al., 2019).

Lastly, we used PyTorch's off-the-shelf implementation of L-BFGS in our experiments (Paszke et al., 2019). While it generally produces good results, we observed some failure cases due to issues while bracketing the step size, leading to `NaNs` or `Infs` and causing the optimization to fail. This would trigger the need for a larger sample size, increasing computational cost. However, a more robust implementation could potentially recover from these failures and continue the optimization process.

## 7 CONCLUSION

In this paper, we introduced the SAA for VI algorithm, which provides an effective and accurate solution to variational inference problems, significantly reducing the reliance on manual hyperparameter tuning. This promising method enhances both efficiency and precision in addressing these challenges.

**Acknowledgements**

This material is based upon work supported by the National Science Foundation under Grant Nos. 1749854, 1908577, and 2045900.

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

# A    DATASETS DESCRIPTION

We use the same datasets as Burroni et al. (2023). The table below, adapted from their paper, provides a summary of the datasets employed.

Table 2: Description of datasets/models.

|  | Num. of variables | Num. of records | Comments |
|---|---|---|---|
| **Bayesian log. regr.** | | | |
| a1a | 105 | 1605 | First 1605 instances of the Adult Data Set, following LIBSVM Chang and Lin (2011), + discretized continous and dummified. |
| australian | 35 | 690 | From UCI + dummified. |
| ionosphere | 35 | 351 | From UCI |
| madelon | 500 | 4400 | From UCI |
| mushrooms | 96 | 8124 | From UCI + dummified. |
| sonar | 61 | 208 | From UCI |
| **Stan models** | | | |
| congress | 4 | 343 | Gelman and Hill (2006) Ch. 7 |
| election88 | 95 | 2015 | Gelman and Hill (2006) Ch. 19 |
| election88Exp | 96 | 2015 | Gelman and Hill (2006) Ch. 19 |
| electric | 100 | 192 | Gelman and Hill (2006) Ch. 23 |
| electric-one-pred | 3 | 192 | Gelman and Hill (2006) Ch. 23 |
| hepatitis | 218 | 288 | WinBUGS Lunn et al. (2000) examples |
| hiv-chr | 173 | 369 | Gelman and Hill (2006) Ch. 7 |
| irt | 501 | 30105 | Gelman and Hill (2006) Ch. 14 |
| mesquite | 3 | 46 | Gelman and Hill (2006) Ch. 4 |
| radon | 88 | 919 | `radon-chr` from Gelman and Hill (2006) Ch. 19 |
| wells | 2 | 3020 | Gelman and Hill (2006) Ch. 7 |

# B    DETAILED COMPARISON WITH ADAM

We provide further details on the experimental setup introduced in Section 5.1. We used the Adam optimizer with the default parameters from the `torch.optim` package in PyTorch (Paszke et al., 2019), with the exception of the step-size, which we varied across 0.1, 0.01, and 0.001. To approximate the distributions, we used a Gaussian with a diagonal covariance matrix and a more expressive Gaussian with a dense covariance matrix. We show the median ELBO values achieved by Adam and SAA for VI in Table 3, and the running time in Table 4. Additionally, we provide the results disaggregated by steps size in Tables 5 and 6. In all instances, we conducted 20 repetitions of the experiments, estimating the objective function with 16 samples from the variational approximation $q_{\theta_t}$. Every 100 iterations, we estimated the ELBO using $10,000$ fresh samples from $q_{\theta_t}$. Although our initial experiments spanned $40,000$ iterations, the dense approximation yielded unsatisfactory results for certain models. Consequently, we extended the number of iterations for these models. Specifically, the `irt` model was run for $200,000$ iterations, while the `madelon`, `election88`, `electric`, and `radon` models were executed for $400,000$ iterations. Despite these extensions, only minor changes in the maximum achieved ELBO were observed. It's noteworthy that the `hepatitis` model diverged when executed beyond $40,000$ iterations using the dense approximation.

| | Diagonal Covariance | | | Dense Covariance | | |
|---|---|---|---|---|---|---|
| | SAA for VI | Adam | Improvement | SAA for VI | Adam | Improvement |
| | (i) | (ii) | (i) − (ii) | (iv) | (v) | (iv) − (v) |
| **Bayesian log. regr.** | | | | | | |
| a1a | -655.51 | -654.79 | -0.72 | -636.40 | -637.23 | 0.83 |
| australian | -269.35 | -268.36 | -0.99 | -256.73 | -256.82 | 0.09 |
| ionosphere | -139.62 | -138.30 | -1.31 | -124.35 | -124.44 | 0.09 |
| madelon | -2,466.15 | -2,466.28 | 0.13 | -2,399.65 | -2,600.32 | 200.67 |
| mushrooms | -211.43 | -210.00 | -1.42 | -179.89 | -180.60 | 0.71 |
| sonar | -151.69 | -149.58 | -2.11 | -110.04 | -110.33 | 0.29 |
| **Stan models** | | | | | | |
| congress | 421.79 | 421.91 | -0.12 | 423.55 | 423.58 | -0.03 |
| election88 | -1,420.01 | -1,419.02 | -0.99 | -1,398.03 | -1,645.18 | 247.15 |
| election88Exp | -1,380.18 | -1,376.03 | -4.15 | -1,381.79 | — | — |
| electric | -788.89 | -788.84 | -0.05 | -786.91 | -859.26 | 72.35 |
| electric-one-pred | -818.36 | -818.33 | -0.03 | -818.01 | -818.00 | 0.01 |
| hepatitis | -560.44 | -560.43 | -0.01 | -557.36 | -618.76 | 61.40 |
| hiv-chr | -608.77 | -608.42 | -0.35 | -582.78 | — | — |
| irt | -15,887.92 | -15,888.03 | 0.11 | -15,884.67 | -15,936.06 | 51.39 |
| mesquite | -30.15 | -30.08 | -0.07 | -29.83 | -29.78 | -0.05 |
| radon | -1,210.70 | -1,210.65 | -0.05 | -1,209.46 | -1,216.92 | 7.46 |
| wells | -2,042.45 | -2,042.37 | -0.08 | -2,041.95 | -2,041.90 | -0.05 |

Table 3: Comparison of SAA for VI and Adam: Median of the highest **ELBO** achieved across multiple optimization runs with different seeds for each model and approximating distribution. Adam was optimized using step sizes of 0.1, 0.01, and 0.001, reporting the configuration with the highest median ELBO. The improvement in median ELBO achieved by SAA for VI over Adam is also included.

## B.1 LARGER SCALE EXPERIMENT

To compare the performance of SAA for VI and Adam on a larger model, we used the `stochastic volatility` model from the Stan library (Carpenter et al., 2017). Following Lai et al. (2022), we modeled the exchange rates of 23 international currencies against the US dollar as stochastic volatilities.[4] To increase the complexity of the task, we employed daily data from 2021/01/01 to 2023/12/31, resulting in a model with $17,228$ latent variables. In Figure 4, we present the ELBO achieved by SAA for VI and Adam over time, showing that SAA for VI reaches and ELBO higher than the optimal ELBO achieved by Adam several times faster.

---

[4]Data can be downloaded from the Federal Reserve.

| | Diagonal Covariance | | | Dense Covariance | | |
|---|---|---|---|---|---|---|
| | SAA for VI | Adam | Improvement | SAA for VI | Adam | Improvement |
| | (i) | (ii) | (ii)/(i) | (iv) | (v) | (v)/(iv) |
| **Bayesian log. regr.** | | | | | | |
| a1a | 0.38 | 18.09 | 48.24 | 19.69 | 19.95 | 1.01 |
| australian | 0.21 | 15.21 | 70.76 | 4.81 | 14.73 | 3.06 |
| ionosphere | 0.17 | 11.44 | 67.64 | 4.33 | 13.47 | 3.11 |
| madelon | 0.82 | 21.02 | 25.62 | 58.52 | 223.55 | 3.82 |
| mushrooms | 0.37 | 27.23 | 73.25 | 17.30 | 29.11 | 1.68 |
| sonar | 0.30 | 11.76 | 39.47 | 12.17 | 11.74 | 0.96 |
| **Stan models** | | | | | | |
| congress | 0.95 | 36.56 | 38.56 | 0.82 | 50.34 | 61.46 |
| election88 | 12.11 | 283.19 | 23.39 | 199.76 | 1,465.89 | 7.34 |
| election88Exp | 12.35 | 261.83 | 21.19 | 83.68 | — | — |
| electric | 1.92 | 65.14 | 33.96 | 42.14 | 235.40 | 5.59 |
| electric-one-pred | 0.51 | 55.22 | 107.75 | 0.62 | 70.62 | 114.40 |
| hepatitis | 2.74 | 103.89 | 37.88 | 96.09 | 264.52 | 2.75 |
| hiv-chr | 2.27 | 56.80 | 24.98 | 29.74 | — | — |
| irt | 1.70 | 33.53 | 19.67 | 94.80 | 210.05 | 2.22 |
| mesquite | 0.73 | 28.87 | 39.47 | 0.27 | 48.54 | 179.91 |
| radon | 1.57 | 74.83 | 47.72 | 18.66 | 252.85 | 13.55 |
| wells | 0.69 | 16.87 | 24.34 | 0.08 | 18.33 | 221.36 |

Table 4: Comparison of **running time**, in seconds, for SAA for VI and Adam across different datasets and distribution approximations, and the ratio of running time improvement of SAA for VI over Adam. Values of ratio greater than 1 indicate that SAA for VI is faster than Adam. SAA for VI generally outperforms Adam, with the exception of the sonar dataset. When using the diagonal covariance approximation, the speed improvement for SAA for VI is notably higher, reaching at least an order of magnitude in most cases. See Section 5.1 for more information.

Table 7: Maximum **ELBO** achieved by Adam and SAA for VI with Gaussian distribution and **diagonal** covariance matrix as approximating distribution: median across seeds. The maximum median ELBO achieved by SAA for VI is higher than the maximum median ELBO achieved by Adam for the stochastic volatility model.

| | Adam—Step Sizes | | | SAA for VI |
|---|---|---|---|---|
| | 0.1 | 0.01 | 0.001 | |
| Stochastic volatility model | 66,532 | 66,811 | 65,770 | 66,845 |

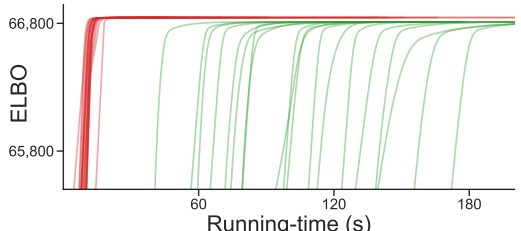

Figure 4: Stochastic volatility model optimized using a diagonal-covariance Gaussian distribution, showing the ELBO achieved by SAA for VI and Adam as a function of time. For Adam, we show the traces corresponding to the best step size of the three used.

| | Adam—Step Size | | | SAA for VI |
|---|---|---|---|---|
| | 0.1 | 0.01 | 0.001 | |
| **Bayesian log. regr.** | | | | |
| a1a | -656.19 | -654.98 | -654.79 | -655.51 |
| australian | -268.85 | -268.42 | -268.36 | -269.35 |
| ionosphere | -138.87 | -138.38 | -138.30 | -139.62 |
| madelon | -2,494.73 | -2,470.07 | -2,466.28 | -2,466.15 |
| mushrooms | -210.97 | -210.22 | -210.00 | -211.43 |
| sonar | -151.09 | -149.80 | -149.58 | -151.69 |
| **Stan models** | | | | |
| congress | 421.86 | 421.90 | 421.91 | 421.79 |
| election88 | -1,436.20 | -1,420.16 | -1,419.02 | -1,420.01 |
| election88Exp | -1,376.35 | -1,376.03 | -1,381.95 | -1,380.18 |
| electric | -790.66 | -789.06 | -788.84 | -788.89 |
| electric-one-pred | -818.34 | -818.33 | -1,063.98 | -818.36 |
| hepatitis | -564.05 | -560.83 | -560.43 | -560.44 |
| hiv-chr | -611.75 | -608.82 | -608.42 | -608.77 |
| irt | -15,896.00 | -15,889.39 | -15,888.03 | -15,887.92 |
| mesquite | -30.09 | -30.08 | -30.08 | -30.15 |
| radon | -1,211.57 | -1,210.79 | -1,210.65 | -1,210.70 |
| wells | -2,042.38 | -2,042.37 | -2,042.37 | -2,042.45 |

Table 5: Maximum **ELBO** achieved by Adam and SAA for VI with Gaussian distribution and **diagonal** covariance matrix as approximating distribution: median across seeds. The table shows the median of the maximum ELBO achieved by Adam and SAA for each model when using a Gaussian distribution with diagonal covariance matrix as approximating distribution. For each step-size used with Adam, we ran the algorithm 20 times and reported the median of the maximum ELBO achieved.

| | Adam—Step Sizes | | | SAA for VI |
|---|---|---|---|---|
| | 0.1 | 0.01 | 0.001 | |
| **Bayesian log. regr.** | | | | |
| a1a | -1,355.11 | -646.20 | -637.23 | -636.40 |
| australian | -269.97 | -257.53 | -256.82 | -256.73 |
| ionosphere | -148.71 | -125.21 | -124.44 | -124.35 |
| madelon | -66,648.98 | -7,599.58 | -2,600.32 | -2,399.65 |
| mushrooms | -242.99 | -182.65 | -180.60 | -179.89 |
| sonar | -386.12 | -114.58 | -110.33 | -110.04 |
| **Stan models** | | | | |
| congress | 423.36 | 423.53 | 423.58 | 423.55 |
| election88 | — | -1,645.18 | — | -1,398.03 |
| election88Exp | — | — | — | -1,381.79 |
| electric | — | -859.26 | — | -786.91 |
| electric-one-pred | -818.01 | -818.00 | -1,083.04 | -818.01 |
| hepatitis | — | -618.76 | — | -557.36 |
| hiv-chr | — | — | — | -582.78 |
| irt | -126,355.62 | -18,773.00 | -15,936.06 | -15,884.67 |
| mesquite | -29.80 | -29.79 | -29.78 | -29.83 |
| radon | — | -1,216.92 | -43,570.33 | -1,209.46 |
| wells | -2,041.91 | -2,041.90 | -2,041.90 | -2,041.95 |

Table 6: Maximum **ELBO** achieved by Adam and SAA for VI with Gaussian distribution and **dense** covariance matrix as approximating distribution: median across seeds. The table shows the median of the maximum ELBO achieved by Adam and SAA for each model when using a gaussian distribution with dense covariance matrix as approximating distribution. For each step-size used with Adam, we ran the algorithm 20 times and reported the median of the maximum ELBO achieved.

## C   DETAILED COMPARISON WITH BATCHED QUASI-NEWTON

In this section, we provide further details regarding experiments conducted using the batched quasi-Newton method as described by Liu and Owen (2021). We compare the maximum ELBO attained by the batched quasi-Newton to that achieved by SAA for VI. This comparison is made for both the Gaussian distribution with a diagonal covariance matrix (Table 8) and the one with a dense covariance matrix (Table 9). While results in the diagonal scenario align closely with ours, the batched quasi-Newton method often converges to a suboptimal solution in the dense case.

Additionally, we report the wall-clock time for each experiment in Table 10. We executed each experiment for 40,000 iterations and performed 20 independent runs for each one. Our method incorporates a stopping criterion based on convergence. To ensure a fair comparison with batched quasi-Newton, we need to detect when the algorithm converges. To approximate this, we first calculate the highest ELBO for each of the 20 independent runs using both batched quasi-Newton and SAA for VI. Then, we compute the median ELBO value across the repetitions for each method. Finally, we determine the minimum median ELBO value between the two methods and calculate the total time taken until the algorithm reaches within 1 nat of this minimum median ELBO value. These results are presented in Table 10.

Similar to the experiments with Adam, this calculation does not account for the time spent on sample sizes that were not useful.

| | Diagonal Gaussian | |
| --- | --- | --- |
| | Batched quasi-Newton 16 | SAAforVI |
| **Bayesian log. regr.** | | |
| a1a | $-654.94$ | $-655.51$ |
| australian | $-268.47$ | $-269.35$ |
| ionosphere | $-138.49$ | $-139.62$ |
| madelon | $-2466.58$ | $-2466.15$ |
| mushrooms | $-210.26$ | $-211.43$ |
| sonar | $-150.14$ | $-151.69$ |
| **Stan models** | | |
| congress | $421.91$ | $421.79$ |
| election88 | $-1426.01$ | $-1420.01$ |
| election88Exp | $-1382.64$ | $-1380.18$ |
| electric | $-788.89$ | $-788.89$ |
| electric-one-pred | $-818.33$ | $-818.36$ |
| hepatitis | $-560.58$ | $-560.44$ |
| hiv-chr | $-608.58$ | $-608.77$ |
| irt | $-15\,888.14$ | $-15\,887.92$ |
| mesquite | $-30.08$ | $-30.15$ |
| radon | $-1210.73$ | $-1210.70$ |
| wells | $-2042.37$ | $-2042.45$ |

Table 8: Comparison of the **ELBOs** obtained by batched quasi-Newton and SAA for VI when using a diagonal Gaussian distribution as the approximating distribution. The batched quasi-Newton method of Liu and Owen (2021) is executed using a sample size of 16. Median results are reported from 20 independent runs for each model. The corresponding results for SAA for VI can also be found in column (ii) of Table 3.

| | | Dense Covariance | | | | |
|---|---|---|---|---|---|---|
| | | Batched quasi-Newton—Sample Size | | | | SAA for VI |
| | | 16 | 32 | 64 | 128 | |
| **Bayesian log. regr.** | | | | | | |
| a1a | | -636.49 | | | | -636.40 |
| australian | | -256.80 | | | | -256.73 |
| ionosphere | | -124.44 | | | | -124.35 |
| madelon | ✗ | -2,418.04 | -2,412.23 | -2,407.44 | -2,406.27 | -2,399.65 |
| mushrooms | | -179.96 | | | | -179.89 |
| sonar | | -110.09 | | | | -110.04 |
| **Stan models** | | | | | | |
| congress | | 423.59 | | | | 423.55 |
| election88 | ✗ | $-1.15 \times 10^{12}$ | $-8.26 \times 10^{11}$ | $-7.23 \times 10^{11}$ | $-5.87 \times 10^{11}$ | -1,398.03 |
| election88Exp | ✗ | $-3.47 \times 10^{19}$ | $-1.15 \times 10^{18}$ | $-3.72 \times 10^{16}$ | $-1.86 \times 10^{16}$ | -1,381.79 |
| electric | ✗ | $-5.44 \times 10^{10}$ | $-6.20 \times 10^{9}$ | $-5.05 \times 10^{9}$ | $-6.08 \times 10^{9}$ | -786.91 |
| electric-one-pred | | -1,145.79 | -818.00 | | | -818.01 |
| hepatitis | ✗ | $-1.99 \times 10^{10}$ | $-1.03 \times 10^{10}$ | $-9.56 \times 10^{9}$ | $-1.64 \times 10^{10}$ | -557.36 |
| hiv-chr | ✗ | $-6.44 \times 10^{15}$ | $-1.47 \times 10^{16}$ | $-3.59 \times 10^{15}$ | $-1.87 \times 10^{15}$ | -582.78 |
| irt | ✗ | -20,481.68 | -18,573.30 | -17,263.15 | -16,099.44 | -15,884.67 |
| mesquite | | -29.78 | | | | -29.83 |
| radon | | $-1.58 \times 10^{6}$ | $-5.50 \times 10^{5}$ | -4,473.35 | -1,209.47 | -1,209.46 |
| wells | | -2,041.90 | | | | -2,041.95 |

Table 9: Final **ELBO** by the batched quasi-Newton method for VI using a Gaussian distribution with a dense covariance matrix (Liu and Owen, 2021). The results for SAA for VI are included as a benchmark (column (v) of Table 3). The batched quasi-Newton method frequently converges to suboptimal solutions, indicated by ✗, especially in models from the Stan examples repository. In models like `election88`, the SAA for VI method demonstrates a significant performance advantage. The initial sample size for the batched quasi-Newton method was set to 16 and increased when necessary to enhance the method's ELBO.

| | Diagonal Covariance | | | Dense Covariance | | |
|---|---|---|---|---|---|---|
| | SAA for VI | Batched quasi-Newton | Improvement | SAA for VI | Batched quasi-Newton | Improvement |
| | (i) | (ii) | (i)/(ii) | (iv) | (v) | (iv)/(v) |
| **Bayesian log. regr.** | | | | | | |
| a1a | 0.38 | 2.10 | 5.60 | 20.31 | 8.40 | 0.41 |
| australian | 0.21 | 1.08 | 5.03 | 4.81 | 2.55 | 0.53 |
| ionosphere | 0.17 | 1.10 | 6.50 | 4.33 | 2.35 | 0.54 |
| madelon | 0.81 | 7.82 | 9.71 | 62.98 | 384.02 | 6.10 |
| mushrooms ✗ | 0.37 | 2.26 | 6.07 | 18.84 | 7.31 | 0.39 |
| sonar | 0.30 | 1.28 | 4.28 | 12.48 | 3.72 | 0.30 |
| **Stan models** | | | | | | |
| congress | 0.95 | 2.93 | 3.08 | 0.82 | 4.99 | 6.10 |
| election88 ✗ | 8.96 | 1,660.06 | 185.34 | — | — | — |
| election88Exp ✗ | 9.75 | 799.40 | 82.02 | — | — | — |
| electric ✗ | 1.92 | 18.35 | 9.57 | — | — | — |
| electric-one-pred | 0.51 | 3.45 | 6.73 | 0.62 | 4.53 | 7.33 |
| hepatitis ✗ | 2.74 | 22.29 | 8.13 | — | — | — |
| hiv-chr ✗ | 2.27 | 30.57 | 13.44 | — | — | — |
| irt ✗ | 1.70 | 37.66 | 22.09 | 89.94 | 663.15 | 7.37 |
| mesquite | 0.73 | 1.39 | 1.90 | 0.27 | 0.95 | 3.51 |
| radon | 1.57 | 9.80 | 6.25 | 22.06 | 648.76 | 29.41 |
| wells | 0.69 | 1.04 | 1.49 | 0.08 | 0.50 | 6.08 |

Table 10: Comparison of **running times**, in seconds, for reaching within 1 nat of the minimum median ELBO value between SAA for VI and batched quasi-Newton across various models and approximating distributions. The analysis for the approximation using a dense covariance matrix considers runs with a batch size of 128 for batched quasi-Newton. For models marked with ✗, indicating failure of batched quasi-Newton in the dense covariance matrix approximation, reports are limited to `madelon` and `irt` as they closely approach the maximum ELBO. The table also shows the running time improvement of SAA for VI over batched quasi-Newton; values greater than 1 imply that SAA for VI is faster.

# D   ADDITIONAL RESULTS FOR SAA FOR VI

Table 4 shows the median time taken by SAA for VI to reach the maximum ELBO achieved by Adam. In this section, we present the total time taken by SAA for VI until its completion. Notably, for some models like `election88`, SAA reached an ELBO over 200 nats higher than Adam, clarifying the discrepancies between Table 11 and Table 4. Additionally, we show the ablation study results regarding parameter initialization using a warm start in Table 12.

|  | Diagonal | | Dense | |
| --- | --- | --- | --- | --- |
|  | time | max. size | time | max. size |
| **Bayesian log. regr.** | | | | |
| a1a | 0.46 | $2^8$ | 52.99 | $2^{18}$ |
| australian | 0.22 | $2^6$ | 9.69 | $2^{17}$ |
| ionosphere | 0.16 | $2^6$ | 6.27 | $2^{16}$ |
| madelon | 1.11 | $2^{11}$ | 100.19 | $2^{18}$ |
| mushrooms | 0.42 | $2^8$ | 90.65 | $2^{17}$ |
| sonar | 0.29 | $2^8$ | 19.24 | $2^{18}$ |
| **Stan models** | | | | |
| congress | 0.95 | $2^5$ | 1.10 | $2^8$ |
| election88 | 12.84 | $2^8$ | 264.98 | $2^{15}$ |
| election88Exp | 11.65 | $2^{10}$ | 351.63 | $2^{12}$ |
| electric | 2.41 | $2^{11}$ | 70.07 | $2^{18}$ |
| electric-one-pred | 0.51 | $2^8$ | 0.62 | $2^7$ |
| hepatitis | 3.49 | $2^{12}$ | 163.19 | $2^{18}$ |
| hiv-chr | 2.68 | $2^9$ | 64.87 | $2^{18}$ |
| irt | 13.83 | $2^{14}$ | 473.77 | $2^{18}$ |
| mesquite | 0.73 | $2^5$ | 0.38 | $2^6$ |
| radon | 2.08 | $2^{11}$ | 53.62 | $2^{18}$ |
| wells | 0.70 | $2^5$ | 0.09 | $2^5$ |

Table 11: Median **running time** (in seconds) and corresponding median **sample size** at which convergence occurs for SAA for VI across runs. As described in Section 5, the sample size is limited to a maximum of $2^{18}$, which proved sufficient for all models.

|  | Fresh/Warm time ratio | |
| --- | --- | --- |
|  | Diagonal | Dense |
| **Bayesian log. regr.** | | |
| a1a | 1.11 | 1.78 |
| australian | 1.01 | 1.58 |
| ionosphere | 0.94 | 1.26 |
| madelon | 1.63 | 1.73 |
| mushrooms | 1.31 | 2.04 |
| sonar | 1.12 | 1.44 |
| **Stan models** | | |
| congress | 1.14 | 3.07 |
| election88 | 3.11 | 20.63 |
| election88Exp | 2.16 | 2.59 |
| electric | 2.64 | 4.69 |
| electric-one-pred | 1.05 | 0.75 |
| hepatitis | 2.77 | 2.03 |
| hiv-chr | 2.10 | 2.70 |
| irt | 3.63 | 6.56 |
| mesquite | 0.98 | 1.31 |
| radon | 2.29 | 5.35 |
| wells | 0.96 | 0.99 |

Table 12: **Time ratio** for the fresh start compared to the warm start. Values greater than 1 indicate that using warm start is faster. For the **ELBO**, significant differences ($> 0.1$) were observed only for the `election88` and `election88Exp` models: $-1.77$ and $-3.46$, respectively, with diagonal covariance, and $1.66$ and $3.43$ with dense covariance. Our results suggest that warm start approaches often reduce optimization time.

# E   ADDENDUM

As mentioned in the related work section, a result by Giordano et al. (2023) demonstrates the futility of using a sample size smaller than the dimension of the latent space for the ELBO optimization problem. In this section, we provide a proof sketch of this result, adapted to our notation.

**Theorem E.1** (Theorem 2 of Giordano et al. (2023)). *Let $q_\theta$ be a Gaussian distribution with parameters $\theta = (\mu, LL^{\mathrm{T}})$, where $\mu \in \mathbb{R}^{d_z}$ and $L \in \mathbb{R}^{d_z \times d_z}$ is a lower-triangular matrix with positive diagonal elements. If we draw a sample of size*

$n < d_Z$ *from* $q_{\text{base}}$*, denoted by* $\boldsymbol{\epsilon} = \epsilon_1, \ldots, \epsilon_n$*, then the optimization problem in Eq. (5) is unbounded:*

$$\sup_{\theta \in \Theta} \hat{\mathcal{L}}_{\boldsymbol{\epsilon}}(\theta) = \sup_{\theta \in \Theta} \frac{1}{n} \sum_{i=1}^{n} [\ln p(z_\theta(\epsilon_i), x) - \ln q_\theta(z_\theta(\epsilon_i))] = \infty.$$

*Proof.* Since $n < d_Z$, there exists a nonzero vector $\mathbf{v} \in \mathbb{R}^{d_Z}$ such that $\langle \mathbf{v}, \epsilon_i \rangle = 0$ for all $1 \leq i \leq n$. Without loss of generality, assume that the largest index $\ell$ with $\mathbf{v}_\ell \neq 0$ satisfies $\mathbf{v}_\ell = 1$. Define the lower triangular matrix

$$L_\lambda = \begin{pmatrix} I_{\ell-1} & & \mathbf{0} \\ & \lambda \mathbf{v}^{\mathrm{T}} & \\ \mathbf{0} & & I_{d_Z - \ell}. \end{pmatrix}.$$

Then, we have $(L_\lambda \epsilon_i)_\ell = 0 = (L_0 \epsilon_i)_\ell$ for all $1 \leq i \leq n$. Let $\theta_\lambda = (\mathbf{0}, L_\lambda L_\lambda^{\mathrm{T}})$. For $\lambda > 0$, we obtain

$$\hat{\mathcal{L}}_{\boldsymbol{\epsilon}}(\mathbf{0}, L_\lambda L_\lambda^{\mathrm{T}}) = \frac{1}{n} \sum_{i=1}^{n} [\ln p(L_\lambda \epsilon_i, x) - \ln q_{\theta_\lambda}(L_\lambda \epsilon_i)] = \frac{1}{n} \sum_{i=1}^{n} [\ln p(L_0 \epsilon_i, x) - \ln q_{\theta_\lambda}(L_\lambda \epsilon_i)] = c + \ln \lambda,$$

where $c$ is a constant independent of $\lambda$.

The result follows by letting $\lambda \to \infty$. $\qquad \square$

With this result in mind, we decided to adapt the SAA for VI algorithm by, in the case of a dense covariance matrix approximation, drawing a sample of size $n$, set as twice the smallest power of two exceeding the latent space dimension $d_Z$. Table 13 and 14 present the experimental results alongside the previously computed results. As observed, starting with a larger sample size allows us to reduce the number of iterations required to achieve a certain accuracy. Furthermore, this reduction is substantial. This outcome was anticipated because, when the problem was unbounded, the optimization process for smaller $n$ typically concluded when the maximum number of iterations was reached, meaning the entire computational budget was utilized.

| | Adam | SAA for VI original, min $n = 32$ | | | SAA for VI new, min $n > d$ | | |
|---|---|---|---|---|---|---|---|
| | Time (i) | Min $n$ | Time (ii) | Improvement (i)/(ii) | Min $n$ | Time (iii) | Improvement (i)/(iii) |
| **Bayesian log. regr.** | | | | | | | |
| a1a | 19.95 | 32 | 19.69 | 1.01 | 256 | 4.69 | 4.26 |
| australian | 14.73 | 32 | 4.81 | 3.06 | 128 | 1.14 | 12.96 |
| ionosphere | 13.47 | 32 | 4.33 | 3.11 | 128 | 0.80 | 16.85 |
| madelon | 223.55 | 32 | 58.52 | 3.82 | 1,024 | 2.57 | 86.90 |
| mushrooms | 29.11 | 32 | 17.30 | 1.68 | 256 | 4.43 | 6.57 |
| sonar | 11.74 | 32 | 12.17 | 0.96 | 128 | 2.75 | 4.27 |
| **Stan models** | | | | | | | |
| congress | 50.34 | 32 | 0.82 | 61.46 | 32 | 0.78 | 64.40 |
| election88 | 1,465.89 | 32 | 199.76 | 7.34 | 256 | 45.72 | 32.06 |
| election88Exp | — | 32 | 83.68 | — | 256 | 5.59 | — |
| electric | 235.40 | 32 | 42.14 | 5.59 | 256 | 13.27 | 17.74 |
| electric-one-pred | 70.62 | 32 | 0.62 | 114.40 | 32 | 0.60 | 117.46 |
| hepatitis | 264.52 | 32 | 96.09 | 2.75 | 512 | 11.49 | 23.02 |
| hiv-chr | — | 32 | 29.74 | — | 512 | 4.11 | — |
| irt | 210.05 | 32 | 94.80 | 2.22 | 1,024 | 15.38 | 13.65 |
| mesquite | 48.54 | 32 | 0.27 | 179.91 | 32 | 0.26 | 185.76 |
| radon | 252.85 | 32 | 18.66 | 13.55 | 256 | 7.43 | 34.03 |
| wells | 18.33 | 32 | 0.08 | 221.36 | 32 | 0.08 | 232.47 |

Table 13: Comparison of **running time**, in seconds, for Adam and SAA for VI across various datasets, using a Gaussian approximating distribution with a dense covariance matrix, showing the running time improvement of SAA for VI over Adam. The **minimum sample size** $n$ for SAA in VI is also displayed. We consider two settings: one where the minimum $n$ is set to 32 for all datasets, which corresponds to the configuration used in this paper [cf. Table 4], and another where the minimum sample size is chosen as the nearest power of 2 to twice $d_Z$, the dimension of the latent space. The results indicate that by avoiding the use of small sample sizes, the running time of SAA in VI can be significantly reduced.

| | | Batched quasi-Newton Time (i) | SAA for VI original, min $n = 32$ | | | SAA for VI new, min $n > d$ | | |
|---|---|---|---|---|---|---|---|---|
| | | | Min $n$ | Time (ii) | Improvement (i)/(ii) | Min $n$ | Time (iii) | Improvement (i)/(iii) |
| **Bayesian log. regr.** | | | | | | | | |
| a1a | | 8.40 | 32 | 20.31 | 0.41 | 256 | 5.32 | 1.58 |
| australian | | 2.55 | 32 | 4.81 | 0.53 | 128 | 1.14 | 2.24 |
| ionosphere | | 2.35 | 32 | 4.33 | 0.54 | 128 | 0.80 | 2.93 |
| madelon | | 384.02 | 32 | 62.98 | 6.10 | 1,024 | 7.22 | 53.22 |
| mushrooms | ✗ | 7.31 | 32 | 18.84 | 0.39 | 256 | 5.94 | 1.23 |
| sonar | | 3.72 | 32 | 12.48 | 0.30 | 128 | 2.95 | 1.26 |
| **Stan models** | | | | | | | | |
| congress | | 4.99 | 32 | 0.82 | 6.10 | 32 | 0.78 | 6.39 |
| election88 | ✗ | | | | | | | |
| election88Exp | ✗ | | | | | | | |
| electric | ✗ | | | | | | | |
| electric-one-pred | | 4.53 | 32 | 0.62 | 7.33 | 32 | 0.60 | 7.53 |
| hepatitis | ✗ | | | | | | | |
| hiv-chr | ✗ | | | | | | | |
| irt | ✗ | 663.15 | 32 | 89.94 | 7.37 | 1,024 | 7.24 | 91.55 |
| mesquite | | 0.95 | 32 | 0.27 | 3.51 | 32 | 0.26 | 3.63 |
| radon | | 648.76 | 32 | 22.06 | 29.41 | 256 | 10.67 | 60.78 |
| wells | | 0.50 | 32 | 0.08 | 6.08 | 32 | 0.08 | 6.38 |

Table 14: Comparison of **running time**, in seconds, for batched quasi-Newton and SAA for VI across various datasets, using a Gaussian approximating distribution with a dense covariance matrix, showing the running time improvement of SAA for VI over batched quasi-Newton. The **minimum sample size** $n$ for SAA in VI is displayed. For models where the batched quasi-Newton method did not fully converge (✗), we only show results for mushrooms and irt, as the others diverged. Two settings are considered: one with a minimum $n$ of 32 for all datasets (used in this paper [cf. Table 10]), and another with the minimum sample size set to the nearest power of 2 greater than twice $d_Z$, the dimension of the latent space. As in Table 13, the results indicate that avoiding small sample sizes can significantly reduce the running time of SAA in VI.

# F HYPERPARAMETERS

As with any optimization algorithm, our implementation of the SAA for VI algorithm uses certain constants and hyperparameters. Table 15 details the purpose of each such number, along with the rationale behind our chosen values. We emphasize that SAA for VI performs well across many models without tuning these parameters (our experiments used a single setting): many can be considered constants, while others control tradeoffs between computation and precision in a straightforward way, such as tolerance parameters. While the current hyperparameter values are not tuned, we are open to the possibility of further enhancing the algorithm's performance through careful tuning.

The sequence of sample sizes is controlled by the first two hyperparameters. We tested a variety of exponentially increasing sequences and determined that the performance was largely unaffected by the specific choice. However, the initial sample size showed a more pronounced effect on performance as it could potentially 'save work' by avoiding smaller sample sizes if larger ones are required. This is not always predictable; our addendum, following Giordano et al. (2023)'s concurrent work, refines SAA for VI by tuning this value based on the model and approximation family.

The remaining hyperparameters, listed last in the table, mainly dictate when to halt the process. For example, a user may deem being 1 nat away from the optimum as adequate, thus setting $\delta$ to 1 instead of 0.01. The $\alpha$ (significance level for t-test) could also be adjusted depending on the desired balance between computation cost and approximation precision. Similar parameters are used in most implementations of other optimization algorithms (maximum iterations, absolute/relative tolerance, etc.) and tend to be less critical than parameters like step sizes as they affect the trade-off between computational time and numerical precision rather than the fundamental operation of the algorithm.

| Hyperparameter | Value | Purpose | Justification |
|---|---|---|---|
| Initial sample size ($n$) | 32 | Sets the starting point for the sample size sequence | Arbitrary choice. Refined based on the work of Giordano et al. (2023) in the addendum. |
| Sample size and max iterations sequence ($2n, 2\tau$) | 2 | Determines progression of sample sizes and max inner optimizer iterations | Arbitrary. We tested alternative sequences with negligible performance impact |
| ELBO difference threshold ($\delta$) | 0.01 | Convergence criterion for the optimizer | Conservative choice ensuring precision |
| Max. number of SAA steps (max_t) or max. sample size | $2^{18}$ | Limits total number of SAA steps or sample size | Chosen to ensure optimization usually concludes for other reasons |
| Inner optimizer early exit count (count < 3) | 3 | Specifies how many times inner optimizer can finish after few iterations | We found empirically that this counter was necessary, but we didn't explore other alternatives. |
| VERY_SMALL_ITER for inner optimizer | 5 | Defines what is considered a small number of iterations for the inner optimizer | Arbitrary choice. It is related to the early exit count. |
| Significance level ($\alpha$) for t-test | 1% | Statistical significance criterion | Standard value in significance testing |
| Test set sample size | 10k | Size of the sample set for ELBO estimation | Arbitrary. It is related to $\alpha$ |
| Initial maximum number of iterations for inner optimizer ($\tau$) | 300 | Sets an initial limit for optimizer iterations | Arbitrary. However, it self-adjusts as needed |

Table 15: Hyperparameter choices for our SAA for VI experiments

# G ADDITIONAL EXPERIMENTS WITH ADAM AND ADAGRAD

This section provides supplementary experimental findings with Adam and AdaGrad. We further explore the performance of Adam with two additional sample sizes: 1 and 256. For AdaGrad, we maintain the sample sizes consistent with those discussed in the main body of the text. The step-size search across 0.1, 0.01, and 0.001 remains unchanged in all experiments.

As the sample size increases, the maximum ELBO value in most models tends towards the one obtained using SAA for VI, as demonstrated in Table 16. (We do not show the results for Adam with $n = 1$ because those were of poorer quality than the results for $n = 16$.) Despite this improvement, some models still exhibit significant disparity. It is important to note that Adam's computational cost continues to be higher than SAA for VI, as evidenced by Table 17. Note that the same instances of SAA for VI were used in all scenarios. Meaning, for each SAA for VI iteration, we ran Adam nine times. This repetition is not reflected in the presented numbers.

In AdaGrad's case, as shown in the Tables, there are promising results for Bayesian logistic regression models. However, the same performance does not extend to the Stan models. Only in the `wells` model does the maximum ELBO value closely match that of SAA for VI.

| | SAA for VI | AdaGrad | Adam | Improvements | |
| | | (n=16) | (n=256) | SAA - AdaGrad | SAA - Adam |
| | (i) | (ii) | (iii) | (i) − (ii) | (i) − (iii) |
|---|---|---|---|---|---|
| **Bayesian log. regr.** | | | | | |
| a1a | -636.40 | -636.76 | -636.57 | 0.36 | 0.17 |
| australian | -256.73 | -256.77 | -256.75 | 0.04 | 0.02 |
| ionosphere | -124.35 | -124.39 | -124.36 | 0.04 | 0.01 |
| madelon | -2,399.65 | -2,469.59 | -2,433.07 | 69.94 | 33.42 |
| mushrooms | -179.89 | -181.48 | -180.02 | 1.59 | 0.13 |
| sonar | -110.04 | -110.19 | -110.09 | 0.15 | 0.05 |
| **Stan models** | | | | | |
| congress | 423.55 | 413.88 | 423.59 | 9.67 | -0.04 |
| election88 | -1,398.03 | — | -1,446.37 | — | 48.34 |
| election88Exp | -1,381.79 | — | — | — | — |
| electric | -786.91 | — | -792.28 | — | 5.37 |
| electric-one-pred | -818.01 | -5,572.18 | -818.00 | 4,754.17 | -0.01 |
| hepatitis | -557.36 | — | -566.51 | — | 9.15 |
| hiv-chr | -582.78 | — | -77,190.31 | — | 76,607.53 |
| irt | -15,884.67 | -15,900.00 | -15,894.76 | 15.33 | 10.09 |
| mesquite | -29.83 | -75.93 | -29.78 | 46.10 | -0.05 |
| radon | -1,209.46 | — | -1,210.36 | — | 0.90 |
| wells | -2,041.95 | -2,041.90 | -2,041.90 | -0.05 | -0.05 |

Table 16: Comparison of AdaGrad and Adam to SAA for VI: Median of the highest **ELBO**.

# H  STATISTICAL TEST ABLATION

Algorithm 2 employs a statistical test to decide whether to continue or stop training. Specifically, training continues as long as the means of the log-weights used for training and a new set of log-weights—i.e., an estimation of ELBO—are statistically different. An alternative approach would be to compare the distributions of both training and testing log-weights using tests designed for this task. To examine this, we conducted experiments similar to those in Section 5, comparing distributions with the *two-sample Kolmogorov-Smirnov test* (KS-test) and the *two-sample Cramér-von Mises test* (CvM-test). The findings are detailed in Table 18. Across all cases, the outcomes closely resemble those achieved with the t-test. Generally, the algorithm runs for slightly longer when using the t-test compared to the KS-test or the CvM-test. This delay is attributed to the greater statistical power gained from comparing means rather than distributions. When comparing distributions, the CvM-test yields marginally better results than the KS-test, attributed to the CvM-test's higher statistical power (Stephens, 1974).

| | SAA for VI | AdaGrad | Adam | Improvements | |
|---|---|---|---|---|---|
| | | (n=16) | (n=256) | AdaGrad/SAA | Adam/SAA |
| | (i) | (ii) | (iii) | (ii)/(i) | (iii)/(i) |
| **Bayesian log. regr.** | | | | | |
| a1a | 20.31 | 12.67 | 16.16 | 0.62 | 0.80 |
| australian | 4.81 | 4.74 | 12.94 | 0.99 | 2.69 |
| ionosphere | 4.33 | 2.95 | 12.41 | 0.68 | 2.87 |
| madelon | 60.31 | 85.49 | 17.03 | 1.42 | 0.28 |
| mushrooms | 18.84 | 68.22 | 31.49 | 3.62 | 1.67 |
| sonar | 12.48 | 3.73 | 9.81 | 0.30 | 0.79 |
| **Stan models** | | | | | |
| congress | 0.82 | 18.36 | 39.40 | 22.39 | 48.11 |
| election88 | 200.34 | — | 3,485.68 | — | 17.40 |
| election88Exp | 83.68 | — | — | — | — |
| electric | 49.16 | — | 275.95 | — | 5.61 |
| electric-one-pred | 0.62 | 32.53 | 65.63 | 52.47 | 105.85 |
| hepatitis | 98.69 | — | 250.90 | — | 2.54 |
| hiv-chr | 29.91 | — | 157.79 | — | 5.28 |
| irt | 109.17 | 106.84 | 62.90 | 0.98 | 0.58 |
| mesquite | 0.27 | 39.74 | 39.76 | 147.19 | 147.26 |
| radon | 21.19 | — | 197.21 | — | 9.31 |
| wells | 0.08 | 7.18 | 17.48 | 89.75 | 218.50 |

Table 17: Comparison of **running time**, in seconds, for AdaGrad and Adam to SAA for VI.

|  | ELBO difference (alternative test) − (t-test) | | | |
| --- | --- | --- | --- | --- |
|  | Diagonal Covariance | | Dense Covariance | |
|  | CvM | KS | CvM | KS |
| **Bayesian log. regr.** | | | | |
| a1a | 0.05 | 0.02 | 0.00 | 0.00 |
| australian | −0.01 | −0.46 | −0.00 | −0.00 |
| ionosphere | −0.31 | −0.44 | −0.00 | −0.00 |
| mushrooms | 0.07 | −0.55 | 0.00 | −0.00 |
| madelon | −0.03 | −0.13 | 0.00 | 0.00 |
| sonar | −1.54 | −1.68 | 0.00 | −0.00 |
| **Stan models** | | | | |
| congress | −0.05 | −0.05 | −0.01 | −0.06 |
| election88 | 0.04 | −0.10 | 0.28 | −0.16 |
| election88Exp | −1.13 | −0.89 | −1.85 | −6.35 |
| electric | 0.01 | 0.00 | 0.00 | 0.00 |
| electric-one-pred | 0.00 | 0.00 | 0.00 | 0.00 |
| hepatitis | −0.01 | −0.00 | −0.00 | −0.00 |
| hiv-chr | 0.01 | −0.07 | 0.01 | 0.01 |
| irt | −0.00 | −0.00 | −0.00 | 0.00 |
| mesquite | −0.04 | −0.05 | −0.01 | −0.04 |
| radon | 0.01 | 0.01 | −0.00 | −0.00 |
| wells | 0.00 | −0.00 | −0.00 | −0.01 |

Table 18: Comparison of the performance of the two-sample Kolmogorov-Smirnov test (KS) and the two-sample Cramér-von Mises test (CvM) for the detection of convergence, as alternatives to the t-test, in the experiments of Section 5. The table shows the difference in ELBO when using an alternative test (CvM or KS) instead of the t-test (negative values indicate that a better approximating distribution was found using t-test). From the results, it is clear that there is not much difference in using alternative statistical tests. However, the CvM test appears to be a slightly better replacement for the t-test than the KS test, primarily due to the greater statistical power of the CvM test (Stephens, 1974). This can be observed in the slightly different behavior on the `mushrooms`, `sonar`, and `election88Exp` datasets.