# OpenReview forum: "Sample Average Approximation for Black-Box Variational Inference"
_auai.org/UAI/2024/Conference — UAI 2024 poster_

### Official Review · Reviewer_4Gft · 2024-03-19

**Q2-1 Originality-Novelty:** 2
**Q2-2 Correctness-Technical Quality:** 3
**Q2-5 Clarity Of Writing:** 3

**Q1 Summary And Contributions:**

Motivation: Variational inference seeks to approximate the posterior distribution given some observations by fitting the parameters of a known proposal distribution. Black-Box variational inference seeks to generalize variational inference for arbitrary models. Current methods treat this as a stochastic optimization problem, where at each step of the gradient ascent a noisy estimate of the gradient is used. While simple at the surface, the authors discuss that this approach requires a lot of fine-tuning, i.e. sample sizes, step sizes, choice of gradient estimator. Further they note that second order derivative methods cannot be applied due to the noisiness of the gradient.

Contribution: The authors assert that for reparameterizable domains, we can make use of the sample average approximation algorithm to turn the SOP into a deterministic optimization problem. They further develop an automatic optimization scheme that makes use of the SAA in order to estimate the optimal parameters of q. Here, the authors develop a sequential scheme with a custom stopping criterion. They further show that for this deterministic optimization problem, the use of second-order methods is feasible and can significantly improve performance. Finally, the authors benchmark the performance of this new method against traditional approaches on a range of standard tasks.

**Q2-3 Extent To Which Claims Are Supported By Evidence:**

3: Good: the main claims are supported by convincing evidence (in the form of adequate experimental evaluation, proofs, (pseudo-)code, references, assumptions).

**Q2-4 Reproducibility:**

2: Fair: key resources (e.g. proofs, code, data) are unavailable but key details (e.g. proof sketches, experimental setup) are sufficiently well-described for an expert to confidently reproduce the main results.

**Q3 Main Strengths:**

While a (simultaneously developed) paper on the SAA method for BBVI has been published, this work further formalizes a general algorithmic approach and provides both new ideas in the form of a stopping criterion as well as empirical evidence of its efficacy. The general algorithm is described clearly and is easy to follow. It appears straight-forward to implement and apply to other problems than the ones showcased. The results provide solid empirical evidence of the methods' benefits by showing improvements in performance for a large variety of models.

**Q4 Main Weakness:**

As a paper on SAA for BBVI has recently been published, this work is incremental in nature. The main contribution is the developed algorithmic optimization scheme making use of automatically adapting sample sizes and a custom stopping criterion. However, evidence for the benefits of these contributions is purely empirical. The limitations of this approach or possible future extensions are not discussed.

**Q5 Detailed Comments To The Authors:**

Content:

- Is there any theoretical backing of t-tests being reasonable as a stopping criterion?
- I recommend adding a short discussion of limitations and future work (e.g.: Can this method maybe be extended to models that are not re-parameterizable?)
- No code was provided, if possible I would recommend adding it as a supplement to increase reproducibility

Presentation:

- The main graphic (Figure 1) highlighting the benefit of the SAA for BBVI method is quite confusing: The body of the paper and the caption of the figure describe 11 STAN models. However, 13 points are being shown. It is unclear what they correspond to.
- The structure of the chapters is a bit confusing to follow, perhaps reorganizing would make it a bit more straight-forward
- It is not immediately clear to the reader that this method is limited to the reparameterizable domain

**Q9 Complying With Reviewing Instructions:**

Yes

---

> ### Author Rebuttal · Authors · 2024-04-05
>
> We thank the reviewer for the detailed review and, in particular, for pointing out the mistake in Figure 1. The additional data points you noticed are due to us inadvertently plotting data from all of the models listed in the Appendix, including the logistic regression models, instead of just the Stan models listed in Table 1. We will correct this for the camera-ready version. (Additionally, there are two points that overlap, and the point corresponding to the radon model—showing a 7 nats improvement and being 13 times faster—was unintentionally left out when adding the axis break.)
>
> Thank you for the comment about the reparameterization trick. We will make it clearer early in the paper (in the introduction) that our formulation is limited to reparameterizable families of distributions. As you point out, this is a good topic for future research. It is worth mentioning that a recently posted pre-print by a different team proposed a follow-up method to perform (sequential) SAA for black-box VI that does not rely on the reparameterization trick (albeit optimizing a different objective).
>
> We will also add a section discussing limitations and extensions. Specifically, we will mention two key limitations in addition to the assumption of reparameterizable distribution families. First, the current approach cannot handle very large latent dimensions unless using mean-field approximations. Second, it does not support data subsampling.
> These limitations naturally lead to directions for future research and possible extensions.
> One such extension is the potential use of SAA with optimizers other than quasi-Newton methods that could take advantage of determinism, for example, first-order methods with line search or data subsampling (but without random parameters).
>
> We will also make the code available.
>
> Regarding the paper by Giordano, Ingram, and Broderick titled “Black box variational inference with a deterministic objective: Faster, more accurate, and even more black box,” we wish to clarify that both papers appeared publicly within days of each other (less than five), though theirs made it to print first. Nevertheless, we regard their work as concurrent, and its authors have also publicly acknowledged our paper as such. To put it another way, just a week before our paper became publicly available, there was no documented work proposing the use of SAA for black-box VI.

---

### Official Review · Reviewer_hkNk · 2024-03-21

**Q2-1 Originality-Novelty:** 3
**Q2-2 Correctness-Technical Quality:** 3
**Q2-5 Clarity Of Writing:** 3

**Q1 Summary And Contributions:**

As for the Black box variational inference (BBVI), the difficulty of using stochastic gradient method to solve BBVI optimization problems reliably and robustly limits its applicability. The authors propose a new BBVI optimization method based on sample average approximation (SAA). They have designed an approach called "SAA for VI". They focus on quasi-Newtonian methods, which are well suited to problems with up to hundreds of potential variables.

**Q2-3 Extent To Which Claims Are Supported By Evidence:**

4: Excellent: all claims are supported by very convincing evidence (in the form of comprehensive experimental evaluation, rigorous mathematical proofs, detailed (pseudo-)code, precise references, well-motivated and realistic assumptions) and the authors deliver what they promise.

**Q2-4 Reproducibility:**

3: Good: key resources (e.g. proofs, code, data) are available and key details (e.g. proofs, experimental setup) are sufficiently well-described for competent researchers to confidently reproduce the main results.

**Q3 Main Strengths:**

1)The design of the experiment is reasonable.
2)The experiment of the paper is sufficient.
3)The format of the literature is uniform.
4)The topic of this paper belongs to the forefront of this field.
5)The explanation of relevant work is basically fine.

**Q4 Main Weakness:**

1)The theoretical proof of the paper is a little insufficient.
2) There are a small number of single-line formulas that are not numbered.
3) The conclusion part is less written, and the summary is a little insufficient.

**Q5 Detailed Comments To The Authors:**

1)The theoretical proof of the paper is a little insufficient.
2) There are a small number of single-line formulas that are not numbered.
3) The conclusion part is less written, and the summary is a little insufficient.
4) It is recommended to write a future job outlook.

**Q9 Complying With Reviewing Instructions:**

Yes

---

> ### Author Rebuttal · Authors · 2024-04-05
>
> We thank you for appreciating our work.

---

### Official Review · Reviewer_MnEg · 2024-03-22

**Q2-1 Originality-Novelty:** 1
**Q2-2 Correctness-Technical Quality:** 4
**Q2-5 Clarity Of Writing:** 4

**Q10 Ethical Concerns:**

No.

**Q1 Summary And Contributions:**

The paper proposes an optimization procedure for an evidence lower bound objective in black-box variational inference that is based on an single fixed sample instead of re-sampling. The use of a fixed sample reduces the stochastic optimization problem to a deterministic one, which allows for determining an appropriate step size using line search and the Wolfe conditions. The paper proposes a protocol for automatic selection of optimization hyperparameters and a protocol for checking convergence. The procedure is compared to two common optimizers on a variety of data sets.

**Q2-3 Extent To Which Claims Are Supported By Evidence:**

2: Fair: the main claims are somewhat supported by evidence (but the experimental evaluation may be weak, or does not match entirely with the claims, important baselines may be missing, proofs contain important ideas but lack rigor, algorithmic details are only discussed superficially, references are imprecise, assumptions are not sufficiently motivated or explicated, etc.).

**Q2-4 Reproducibility:**

3: Good: key resources (e.g. proofs, code, data) are available and key details (e.g. proofs, experimental setup) are sufficiently well-described for competent researchers to confidently reproduce the main results.

**Q3 Main Strengths:**

- The paper is well-written.
- The proposed optimization procedure is much faster than the Adam optimizer.

**Q4 Main Weakness:**

- The paper lacks depth. For example, the ablation study is two paragraphs of text and an extensive table but is ultimately not insightful (warm start seems empirically useful sometimes but not always). Another example is the proposed solution to the detection of overfitting when the optimizer finishes in a small number of iterations: just force the optimizer to run a "very_small_iter" number of times.
- Justification for the procedure is largely empirical, with little analysis.
- The convergence check procedure contains arbitrary steps, such as a p-value for t-testing and a convergence threshold \delta, with little exploration of choice effects.

**Q5 Detailed Comments To The Authors:**

- (Sec. 2) What is $C^{1}$? The space of first-order differentiable continuous functions?
- (Figure 3) I don't understand this visualization; the caption mentions ".. methods more than 100 nats worse are not shown". But the best models are normalized to 100 nats, so there are no methods with more nats? Does that imply that methods worse than 100 nats are not shown? In the figure, some methods worse than 100 nats are shown.
- (Sec. 4) It is mentioned that there is a discrepancy between the theoretical results of Giordano et al. (regarding a failure mode of SAA when sample size n is less than dimensionality d_Z) and the experimental results the paper describes. Two reasons are indicated: 1) this paper uses a larger sample size and 2) this paper uses fewer latent variables. I do not see how the experiments contradict the theoretical result: it seems these experiments adhere to the recommendations by Giordano et al.
- (Sec. 5.1) typo "achiaved"

**Q9 Complying With Reviewing Instructions:**

Yes

---

> ### Author Rebuttal · Authors · 2024-04-05
>
> It was not mentioned explicitly, but we infer from the comment “close to other recently presented ideas” in the overall justification and your score of 1 for Q1 (originality-novelty) that you are judging our work relative to that of Giordano, Ingram, and Broderick titled “Black box variational inference with a deterministic objective: Faster, more accurate, and even more black box.” Since that work is concurrent with ours [referenced in the last paragraph of the Introduction section in our submission, page 2], we encourage you to please revisit that mistaken premise. Note that both papers first appeared publicly within days of each other, and both author teams have publicly acknowledged the works as concurrent, despite the fact that theirs has now made it to print. If you are referring to different recently presented ideas, we would appreciate a specific reference.
>
> Regarding several of the weaknesses mentioned in Q4, we would highlight that every optimization procedure when implemented in practice has tuning and tolerance parameters, sometimes a large number of them. A good test of whether a procedure is practically useful is whether it is possible to find default settings for these tuning parameters that work well on a broad range of problems. That has been very hard to do for Adam or SGD applied to BBVI. A key finding of our paper is that SAA can be much more robust (and also faster and more accurate) than Adam of SGD. We consider providing a procedure with good default settings and practical heuristics that work well for many problems to be key evidence for this finding. However, we don’t consider these individual elements to be our main conceptual contributions and did not elaborate on them as such. Rather, our main conceptual contribution was the application of a novel stochastic optimization approach, SAA, to BBVI. Again, perhaps you have judged our work as a follow-on, instead of being concurrent to, Giordano et al.
>
> Regarding the “discrepancy” with the result of Giordano et al: of course we do not contradict the theoretical result. The result is clearly correct, and we have included a statement and self-contained proof of it in our addendum (Appendix E) as context. The discrepancy is with the practical conclusions drawn from the result. We say “they conclude this limitation prevents the use of SAA for VI with a dense Gaussian approximation”, but “we show in the experiments section that, for interesting models, it is indeed feasible”, and then continue to talk about this “discrepancy”. Indeed, we have communicated directly with their author team and they are happy with our characterization of their work.

---

### Official Review · Reviewer_Uuib · 2024-03-22

**Q2-1 Originality-Novelty:** 2
**Q2-2 Correctness-Technical Quality:** 3
**Q2-5 Clarity Of Writing:** 2

**Q1 Summary And Contributions:**

The manuscript introduced a novel optimization strategy using the sample average approximation (SAA) method, allowing for transforming stochastic optimization issues into deterministic ones for solving Black-box variational inference (BBVI) problem. The proposed model solves a series of deterministic problems with incrementally larger sample sizes to address BBVI optimization challenges without the need for problem-specific adjustments. This method particularly leverages quasi-Newton methods, making it suitable for problems involving up to hundreds of latent variables. Experiments demonstrate that "SAA for VI" not only simplifies the variational inference problem but also outpaces existing methods in terms of speed.

**Q2-3 Extent To Which Claims Are Supported By Evidence:**

3: Good: the main claims are supported by convincing evidence (in the form of adequate experimental evaluation, proofs, (pseudo-)code, references, assumptions).

**Q2-4 Reproducibility:**

3: Good: key resources (e.g. proofs, code, data) are available and key details (e.g. proofs, experimental setup) are sufficiently well-described for competent researchers to confidently reproduce the main results.

**Q3 Main Strengths:**

The authors proposed sample average approximation (SAA) to convert the Black-box variational inference (BBVI) into a deterministic optimization. This enables line-search and quasi-Newton methods in the optimization.

The proposed method achieves better performance than existing methods.

**Q4 Main Weakness:**

The concept of Sample Average Approximation (SAA), which transforms stochastic optimization problems into deterministic optimization problems, has been previously introduced. Utilizing line-search and quasi-Newton methods for solving deterministic optimization problems is also well-established. This paper's contribution appears to be the application of these methods to a specific area, namely Black-box Variational Inference (BBVI). As a result, the novelty of the paper seems to be limited.

The proposed method does not seem practical for large datasets, because it does not incorporate batch processing. Estimating the Hessian matrix for problems with a large number of parameters could pose computational challenges and be memory intensive.

To enhance readability and understanding, it would be beneficial for the authors to include more details about their methodology in the paper.

**Q5 Detailed Comments To The Authors:**

I have concerns regarding the scalability of the proposed method when applied to large datasets or models with a large number of parameters. The method has only been tested on smaller datasets, as shown in the appendix. I am particularly interested in how the method's performance compares to that of Adam, especially as the number of model parameters increases.

The distinction between Equations (5) and (4) is not immediately clear. The authors might consider emphasizing that $\epsilon$ is fixed and not re-sampled during the optimization process to clarify this point.

In Section 3.2, the phrase "double the sample size" is confusing. Given that the dataset size is fixed, I'm unsure how the number of samples can be changed. This part of the text requires further clarification.

Despite the explanations in Section 3.2, the purpose of "count" and the rationale behind doubling $n$ and $\tau$ in Algorithm 1 remain unclear to me. More intuitive explanations would greatly aid in understanding these concepts.

**Q9 Complying With Reviewing Instructions:**

Yes

---

> ### Author Rebuttal · Authors · 2024-04-05
>
> We thank the reviewer for the reviewer's comments and suggestions.
> We present some additional details below to address the reviewer's questions and concerns.
>
> Regarding scalability to larger dimensions, there are two different potential issues. One is the size of the optimization variable $\theta$, which is the number of parameters for the approximate posterior $q_\theta(z)$. We think this is what you had in mind, but please let us know if otherwise. In our experiments, we mostly focused on models with up to 500 latent variables (size of $z$), which means up to about 250K parameters (size of $\theta$) with a dense Gaussian distribution. We also showed an example with 15K latent variables and a mean-field Gaussian (~30K parameters). It is absolutely correct that when $\theta$ becomes very high-dimensional, it will be difficult to apply a second-order optimizer, even one that keeps a compressed approximation of the Hessian like L-BFGS, which we use in our experiments. We would highlight two things:
> * First, models with up to 500 latent variables parameters are quite common in practice (for instance, we believe they represent the majority of models used in Stan) and are often not well solved by existing BBVI methods, especially when using dense Gaussian approximating families (~250K parameters). We would certainly like to be able to take advantage of the strengths of second-order optimizers in this regime, and our work enables doing so in a very robust way that was not possible before.
> * Second, it is possible to use different deterministic optimizers within the SAA methodology. We chose to focus on second-order methods specifically because of the advantages they offer, when applicable. However, it would also be possible to use a first-order method, and we would consider this a fruitful direction for future research. For example, by making the problem deterministic, one could easily use gradient descent with line search, or reintroduce (a different form of) stochasticity through data subsampling.
>
> A second possible scalability issue is related to the size of the observed data $x$—for example, in logistic regression this would represent the number of training pairs—and whether the inference approach supports data subsampling for $x$ when $p(x | z)$ has a suitable independence structure. We guess this is not the main issue you had in mind. Briefly, we don’t consider this is a significant issue because, even if $x$ is big, it does not affect the size of the decision variable $\theta$ and we can compute $\log p(x | z)$ and its gradients with respect to $\theta$ in a streaming fashion without a large memory footprint. It would be another interesting future research direction to incorporate data subsampling into SAA, which could be done following the ideas of the second bullet point above by switching to a first order optimizer for the deterministic problem and then reintroducing this (again, different form of) stochasticity.
>
> We hope this clarifies the applicability of the SAA method for black-box VI on large datasets.
> Regarding the phrase "double the sample size," we clarify that it refers to the number of independent and identically distributed samples from the noise distribution used to approximate the expectation in the SAA objective, identified as $n$ in the submission.
> Since we can decide on how many samples to evaluate the model (and for each sample, the model evaluates the likelihood on the complete dataset), we can choose the number of samples to use.
> The rationale behind doubling \(n\) is that the SAA objective is a Monte Carlo approximation of the ELBO, and the approximation gets closer to the actual value as $n$ increases.
> The rationale for doubling $\tau$, the maximum number of iterations allowed for the quasi-Newton method, differs.
> The most important role of $\tau$ is to detect when the quasi-Newton method is not converging. Hence, we set it to a value that allows the method to converge but is not so large as to extend the runtime unnecessarily. However, we added a self-tuning mechanism to accommodate models that need more iterations to converge.
> We thank you again and will ensure to clarify these points, with intuitive explanations, in the final version.

---

### Official Review · Reviewer_dYWw · 2024-03-25

**Q2-1 Originality-Novelty:** 3
**Q2-2 Correctness-Technical Quality:** 2
**Q2-5 Clarity Of Writing:** 3

**Q1 Summary And Contributions:**

The authors propose a new optimisation algorithm for VI that samples a fixed random noise and optimises the objective with plug-in estimate. This results in a deterministic objective that can be optimised using more advanced algorithms.

**Q2-3 Extent To Which Claims Are Supported By Evidence:**

3: Good: the main claims are supported by convincing evidence (in the form of adequate experimental evaluation, proofs, (pseudo-)code, references, assumptions).

**Q2-4 Reproducibility:**

3: Good: key resources (e.g. proofs, code, data) are available and key details (e.g. proofs, experimental setup) are sufficiently well-described for competent researchers to confidently reproduce the main results.

**Q3 Main Strengths:**

I think this idea is novel. To my mind, this technique is effectively a bias variance trade-off taken to the extreme, there's no variance, just bias.

**Q4 Main Weakness:**

Since this approach an be viewed as a bias-variance trade-off, the introduced bias should be somehow quantified.
The discussion in Detection and mitigation of overfitting section should be expanded. Assumption of convergence to global optimum is very limiting.

The discussion of Wolfe conditions can be moved to the appendix which would allow the authors focus more on the contributions.

The differences in Table 1 appear to be very small in some cases.

**Q5 Detailed Comments To The Authors:**

How does the performance depends on the number of used noise samples?

How does this change with dimensionality of a problem?

What happens if you use the same objective (fixed noise samples) with Adam?

Can you explain in more detail what would happen in Gaussian case?

 Are results in Table 1 statistically significant at all? Why the algorithm appears to be working only in some cases?

**Q9 Complying With Reviewing Instructions:**

Yes

---

> ### Author Rebuttal · Authors · 2024-04-05
>
> We appreciate your interest and hope you find our responses useful.
>
> Regarding the SAA bias, which is often considered sampling error due to the approximation of the true objective by a sample average, quite a lot is known in the SAA literature and we will expand our discussion to include more about it. In particular, we will highlight that SAA convergence results do not depend on convergence of the optimization procedures. Rather, they argue about convergence of the SAA objective (Eq. (5)) to the true objective (Eq. (4)) and the convergence of the optima of the SAA problem to those of the true problem as $n\to\infty$.
> Briefly, it’s easy to see by the law of large numbers that the SAA objective, which is a sample average, converges pointwise (at any $\theta$) to the true objective. Then, under various regularity conditions it is possible to argue that the SAA objective converges uniformly to the true objective, which allows arguing that the optimal value and solution(s) of the SAA problem converge to those of the true problem. Under additional regularity conditions, it is possible to establish a $1/\sqrt{n}$ rate of convergence and asymptotic normality for both the optimal value and solution. SAA is a well established methodology and these arguments have been applied in a broad range of settings — please see Section 4 of “A Guide to Sample-Average Approximation" by Kim et al. (2015) for formal statements of some of the known results. Note that we have observed empirically the $1/\sqrt{n}$ convergence rate expected for SAA when applied to BBVI, but have not formally shown that the regularity conditions needed to apply an existing convergence theorem hold for BBVI. We view this as an interesting and potentially challenging direction for future research. It can be compared to the situation of solving the BBVI problem with stochastic gradient (SG) methods: SG is known to converge under a wide range of conditions, and was used with BBVI for nearly a decade before the first formal convergence guarantees were given for SG+BBVI; see Kim, Oh, Wu, Ma, and Gardner (2023) and Domke, Gower, and Garrigos (2023).
>
>
> We would also like to point out that the heuristics that we use for detecting failure of SAA to converge are quite robust to imperfect optimization. For example, suppose the deterministic optimizer searches only a subset $A\subseteq\Theta$ to return the suboptimal solution $\hat\theta\in\arg\max_{\theta \in A}\hat{\cal L}\_\epsilon(\theta)$. It is still true that the  training objective $\hat{\cal L}\_\epsilon(\hat \theta)$ at the suboptimal solution $\hat \theta$ is, in expectation, an upper bound to $\cal L(\hat \theta)$, which are the values we compare. This follows from the same reasoning as Mak et al., using Jensen's inequality:
> $$\def\L{\cal L}\def\Le{\hat{\L}\_\epsilon}\def\E{\mathbb E}\def\mtA{\max\_{\theta \in A}}
> \E[\Le(\hat\theta)]=\E[\mtA \Le(\theta)]\geq\mtA\E[\Le(\theta)]=\mtA\L(\theta)\geq\L(\hat \theta).
> $$
> We only need the deterministic optimizer to converge in the case when we want an upper bound to the true optimal value $\cal{L}(\theta^*)$. In that case, take $A=\Theta$ above, and the second-to-last quantity is $\cal L(\theta^*)$. Finally, even though obtaining an upper bound on the true optimal ELBO $\cal L(\theta^*)$ requires perfect deterministic optimization, we still found it to be very useful empirically. In cases when we believe the deterministic problem is well behaved, we can assess the quality of our solution compared to what is possible given a family of approximating distributions. This feature is very unique to the SAA approach.
>
>
>
> Regarding the results in Table 1, it is somewhat expected that for simple problems, the solutions found by the SAA method for VI are similar to those found by Adam, especially in scenarios where Adam is able to find a global solution.
> For instance, in the case of the smallest model in our collection, the wells model, which consists of two latent variables, when approximating the posterior with a Gaussian distribution that has a dense covariance matrix, Adam finds the location of this Gaussian to be $(0.6030, -0.6189)$, while the covariance matrix is $\begin{pmatrix} 0.0036 & -0.0043\\ -0.0043 &  0.0090\end{pmatrix}$. Similarly, the SAA for VI method locates the Gaussian at $(0.6146, -0.6260)$ with a covariance matrix of $\begin{pmatrix} 0.0032 & -0.0039\\ -0.0039 &  0.0097\end{pmatrix}$.
> Thus, the parameters identified by Adam and SAA are quite close to each other and the difference in the ELBO is minimal.
> However, even in this simple scenario, SAA provides benefits, notably finding the solution in 0.08 seconds compared to the 18.33 seconds required by Adam.
>
> To conclude, we believe that the SAA method is a promising approach for black-box VI, not only because it may find better solutions than Adam in more complex problems, but also because it can provide a good solution in a fraction of the time, while requiring less involvement from the user.

---

### Meta-Review · Area_Chair_BWUp · 2024-04-17

The main point of contention with this work among the reviewers is that this work appeared concurrently with a more theoretically-minded work by different authors.  The key issue that the reviewers appear to have is that this work's theoretical novelty is limited and any claims that it has made over the already published other work are only empirically validated.  This is a tough call in my opinion.  The empirical contributions seem to be worthwhile on their own, but some reviewers felt that additional details of the approach would be needed to make a more thorough evaluation.  Weighing these issues, I believe that the fact that there was a concurrent work only really limits the novelty of the proposed approach (which none of the reviewers seem to have a problem with).  As a result, I tend to think that this work would benefit from additional revision that more carefully articulates the details of the proposed approach.